# Development of crystal orientation fabric in the Dome Fuji ice core in East Antarctica: implications for the deformation regime in ice sheets

Tomotaka Saruya[1], Shuji Fujita[1,2], Yoshinori Iizuka[3], Atsushi Miyamoto[4], Hiroshi Ohno[5], Akira Hori[5], Wataru Shigeyama[2*], Motohiro Hirabayashi[1], Kumiko Goto-Azuma[1,2]

[1] National Institute of Polar Research, Tokyo 190-8518, Japan
[2] Department of Polar Science, The Graduate University for Advanced Studies, SOKENDAI, Tokyo 190-8518, Japan
[3] Institute of Low Temperature Science, Hokkaido University, Sapporo 060-0819, Japan
[4] Institute for the Advancement of Higher Education, Hokkaido University, Sapporo 060-0817, Japan
[5] Kitami Institute of Technology, Kitami 090-8507, Japan

*Currently at: JEOL Ltd., Tokyo 196-8558, Japan

Correspondence: Tomotaka Saruya (saruya.tomotaka@nipr.ac.jp)

**Abstract.** The crystal orientation fabric (COF) of a polar ice sheet has a significant effect on the rheology of the sheet. With the aim of better understanding the deformation regime of ice sheets, the work presented here investigates the COF in the upper 80% of the Dome Fuji Station ice core in East Antarctica. Dielectric anisotropy ($\Delta\varepsilon$) data were acquired as a novel indicator of the vertical clustering of COF resulting from vertical compressional strain within the dome. The $\Delta\varepsilon$ values were found to exhibit a general increase with depth, but with fluctuations over distances on the order of $10$-$10^2$ m. In addition, significant decreases in $\Delta\varepsilon$ were found to be associated with depths corresponding to three major glacial to interglacial transitions. These changes in $\Delta\varepsilon$ are ascribed to variations in the deformational history caused by dislocation motion occurring from near-surface depths to deeper layers. Fluctuations in $\Delta\varepsilon$ over distances of less than 0.5 m exhibited a strong inverse correlation with $\Delta\varepsilon$ at depths greater than approximately 1200 m, indicating that they were enhanced during the glacial/interglacial transitions. The $\Delta\varepsilon$ data also exhibited a positive correlation with the concentration of chloride ions and an inverse correlation with the amount of dust particles in the ice core at greater depths corresponding to decreases in the degree of *c*-axis clustering. Finally, we found that fluctuations in $\Delta\varepsilon$ persisted to approximately 80% of the total depth of the ice sheet. These data suggest that the factors determining the deformation of ice include the concentration of chloride ions and the amount of dust particles, and that the layered contrast associated with the COF is preserved all the way from the near-surface to a depth corresponding to approximately 80% of the thickness of the ice sheet. These findings provide important implications regarding further development of the COF under the various stress-strain configurations that the ice will experience in the deepest region, approximately 20% of the total depth from the ice/bed interface.

# 1. Introduction

The crystal orientation fabric (COF) is one of the most important factors determining the physical properties of polar ice sheets, as both the deformation and flow of ice sheets are highly dependent on the COF. It is commonly accepted that dislocation creep is the dominant deformation process in polar ice sheets (e.g., Cuffey and Paterson, 2010; Petrenko and Whitworth, 1999). In addition, in the dome summit regions of ice sheets, the vertical compressional stress imparted by the mass of the ice is the primary deformation stress. In such cases, the $c$-axes of the ice crystal grains rotate toward the compression direction and the COF becomes more concentrated toward the core axis (that is, in the vertical direction) with increasing depth (e.g., Thorsteinsson et al., 1997; Azuma et al., 2000; Wang et al., 2003; Durand et al., 2007, 2009; Montagnat et al., 2014). Thus, profiling the degree of clustering of the $c$-axes in the depth direction is a useful means of examining the nature of the flow in an ice sheet and evaluating the deformation history. This analysis also allows assessment of further developments in ice flow under simple shear or under more complex stress/strain configurations in the deeper interiors of ice sheets. Therefore, a better understanding of the COF is essential to improve present-day ice flow models as well as to accurately date deep ice cores and predict ice deformation near the base of an ice sheet.

Various studies of ice cores have established the relationships between the microstructure of polycrystalline ice and climate change events such as glacial/interglacial periods and termination events. For example, the grain size in glacial ice is finer than that in interglacial ice (e.g., Paterson, 1991; Thorsteinsson et al., 1995; Cuffey and Paterson, 2010). It has been suggested that the smaller grain sizes in glacial ice result from high concentrations of impurities such as dust particles or soluble substances that restrict grain growth via pinning and drag at the grain boundaries (e.g., Alley et al., 1996; Gow et al., 1997; Durand et al., 2006). However, the COF changes associated with climate stages such as interglacial/glacial periods and termination events are less well understood. As an example, the $c$-axis clustering changes differ in the terminations found in the Dome Fuji (DF), EPICA Dome C (EDC) Antarctica (see Fig. 1 for locations) and GRIP and NEEM Greenland ice cores. Specifically, in the termination II period, the degree of $c$-axis clustering in the DF ice core is reduced (Azuma et al., 2000) while that in the EDC ice core is increased (Durand et al., 2007). In addition, the COF in the termination I portion of the GRIP ice core does not show any observable change (Thorsteinsson et al., 1997) while that in the NEEM ice core exhibits rapid strengthening. These different results are not well understood and highlight a need for more detailed investigations of COF development in ice cores.

For more than two decades, the COF characteristics in various ice domes have been investigated using automated COF analysers to examine thin ice sections (e.g., Azuma et al., 2000; Wang et al., 2003; Durand et al., 2007, 2009; Montagnat et al., 2014). The innovative automated COF analysers used in such studies enabled the rapid assessment of large numbers of crystal grains within each thin section. Still, clear limitations remain because the preparation of numerous thin sections is labour-intensive and therefore significant time and effort are required to obtain a continuous COF profile. Accordingly, the thin section sampling interval has typically been limited to 10–20 m along the ice core (see Table 1 for a detailed comparison of the sampling intervals in each ice core). In addition, the statistical reliability of COF data obtained from thin sections has

yet to be established. As an example, even when evaluating the same samples taken from the EDC Antarctic ice core, two independent groups determined different COF eigenvalues (Wang et al., 2003; Durand et al., 2007, 2009). It is also important to eliminate possible biases and errors resulting from the use of automatic fabric analysers. In short, thin-section-based methods have inherent limitations related to obtaining statistically significant data. Consequently, it has thus far been

challenging to examine small fluctuations in the COF or to compare COF data generated using different algorithms (e.g., Wang and Azuma, 1999; Wilen et al., 2003; Wilson et al., 2003).

To overcome these limitations, Saruya et al. (2022) proposed a technique that permits the continuous non-destructive and rapid assessment of the COF in thick ice sections, based on measuring the tensorial components of the relative permittivity, $\varepsilon$, using microwave open resonators. In this process, the difference in $\varepsilon$ between the vertical and horizontal planes is defined as

the dielectric anisotropy, $\Delta\varepsilon$. Saruya's group demonstrated that $\Delta\varepsilon$ is a direct substitute for the normalized COF eigenvalues when assessing thick sections. Compared to thin-section-based methods, this technique provides COF data with greatly improved statistical significance.

In the present study, we applied this thick-section-based method to an investigation of the COF within an approximately 2300 m long portion of the DF ice core drilled at one of the major dome summits in East Antarctica. The $\Delta\varepsilon$ values in this

sample were measured at 0.02 m intervals in 1 m long ice core specimens acquired every 5 m at depths from 100 to 2400 m. The resulting data were compared with the profiles of various physicochemical parameters, such as major ions, dust particles, salt inclusions and grain sizes, obtained from analyses of the DF ice core. The goal was to better understand the factors influencing COF development. Based on the results, we discuss the possible causes of COF variations, as well as flow mechanism contrast within ice sheets. This paper also discusses the implications for further deformation of the ice in these

locations under specific conditions, including the very deep part of the ice sheet near the ice/bed interface.

**Table 1.**

Comparison of sampling intervals and dimensions (width × height × thickness) for each ice core. The sample width in the present study indicates the half-power diameter of the Gaussian beam. Although the precise thicknesses of thin sections are not provided in Wang et al. (2003) or Durand et al. (2007, 2009) (missing data are indicated by the # symbol in this table), it can be assumed that the thicknesses of sections prepared for optically-based COF measurements were approximately 0.5 mm or less.

| Ice core | Reference | Depth [m] | Sampling interval [m] | Sample dimension [mm] |
|---|---|---|---|---|
| DF2 | This study | 100–2400 | 5 | ~38 × 1000 × 33–79 |
| DF1 | Azuma et al., 2000 | 100–2300 | 20 | 50 × 100 × 0.5 |
| | | 2300–2500 | 10 | |
| EDC | Wang et al., 2003 | 100–1500 | 12–150 | 45 × 90 × # |
| | Durand et al., 2007 | 1500–2000 | 11 | 40 × 110 × # |
| | Durand et al., 2009 | 313–511 | 11 | |
| | | 1500–3100 | 11 | |

## 2. Samples and methods

### 2.1 Sample preparation

This work assessed an ice core drilled at DF, one of the major dome summits in East Antarctica (see Fig. 1), located at 77°19′ S, 39°42′ E, with an elevation of 3800 m. The annual mean temperature at this location is –54.4 °C, the annual accumulation rate is $27.3 \pm 1.5$ kg m$^{-2}$ year$^{-1}$ and the ice thickness is $3028 \pm 15$ m (Dome Fuji Ice Core Project Members, 2017). Figure 1b demonstrates that the ice coring site was very close (within 10 km) to the present dome summit. At present, under the Holocene climate, DF is associated with a steep north-south surface mass balance gradient (Fujita et al., 2011; Tsutaki et al., 2021). We suggest that this morphology demonstrates that the DF summit has migrated along this gradient in the north-south direction during glacial and interglacial periods over which the accumulation rate changed dramatically (e.g., Parrenin et al., 2016). Very deep ice cores were drilled twice at DF. The first 2503 m long core (hereinafter the DF1 ice core) was drilled between 1993 and 1997 (e.g., Watanabe et al., 1999) while the second 3035 m long ice core (hereinafter the DF2 ice core) was drilled between 2004 and 2007 (Motoyama et al., 2007; Dome Fuji Ice Core Project Members, 2017). Note that the DF1 and DF2 boreholes are only 48 m apart. We used the DF2 core in our study, but it should be noted that Azuma et al. (1999, 2000) used a thin-section method to conduct COF studies with the DF1 core. In contrast to this prior study, the

present work employed the thick-section-based method to examine the DF2 core at 5 m intervals between the depths of 100 and 2400 m. At each sampling depth, we continually assessed a 1 m (comprising two 0.5 m sections) long ice core with a 0.02 m step size. Consequently, this work examined approximately 20% of the entire ice core. Each ice core sample was

approximately 0.5 m long and was formed into a slab shape with a thickness of 68–79 mm and width of 53–62 mm. A diagram of the core cutting geometry is shown in Fig. 2a. In the case of specimens acquired between 600 and 870 m, the slab thicknesses were approximately 33–38 mm. Each sample was effectively a cylinder penetrated by the microwave beam having a diameter of 38 mm and a thickness of 33–79 mm. In this study, we focus on COF development within the upper 80% of the ice thickness, meaning depths of up to 2400 m within the 3028 (±15) m thick ice sheet (Fujita et al., 1999). The

age of the ice to a depth of 2400 m was approximately 300 kyrs BP. Note that interpretation of the COF data obtained from dielectric measurements is challenging below 2400 m due to the presence of inclined layers and extremely coarse crystal grains. At these depths, the layered structures begin to incline relative to the horizontal plane, with inclinations of less than 5° above 2400 m but much larger values of 20° and 50° at 2800 and 3000 m, respectively (Dome Fuji Ice Core Project Members, 2017). Additionally, visual inspection of the samples showed extremely large coarse grains (with grain sizes > 50

120    cm) in samples from the deepest part of the dome. The effects of these factors should be confirmed by future research but, for the present, we restricted our analyses to a depth of 2400 m for these reasons. The COF development within the bottom 20% of the ice thickness (from 2400 m to the ice sheet bottom) will be reported elsewhere.

**2.2 Dielectric anisotropy measurements**

The $\Delta\varepsilon$ values for ice cores were determined using an open microwave resonator, employing frequencies between 14 and 20 GHz (Saruya et al., 2022). The operating principle and applications of the open resonator method with regard to obtaining relative permittivity values have been previously described in the literature (Jones, 1976a, b; Cullen, 1983; Komiyama et al., 1991). Using this system, we developed a means of performing continuous measurements of thick slab samples. The present research constructed a semi-confocal type of open resonator incorporating a flat mirror and a concave mirror having a 250

130    mm radius of curvature, set 225 mm apart. A diagram of the experimental setup is provided in Fig. 2b. A microwave beam having a Gaussian profile was generated with a half-power diameter of 38 mm. The $\varepsilon$ values obtained in this work were volume-weighted averages within the volume covered by the Gaussian distribution of the beam. When the angle between the core axis and the electric field was set to 45°, radio birefringence was observed. That is, when the frequency was swept to detect resonances that corresponded to transverse electromagnetic (TEM) $_{0, 0, q}$ modes (where q is an integer), two resonance

peaks resulting from anisotropic permittivity components were detected. The two radio birefringence components corresponded to the $\varepsilon$ values in the horizontal and vertical directions within the core. Because each ice crystal is uniaxially symmetric around its $c$-axis with respect to $\varepsilon$, the degree of $c$-axis clustering around the vertical direction could be evaluated by measuring the macroscopic $\varepsilon$ values both parallel and perpendicular to the ice core axis (e.g., Hargreaves, 1978). In this

work, we measured $\varepsilon$ continuously by moving the ice core sample using an automatic motor. These analyses were conducted at temperatures in the range of $-30 \pm 1.5$ °C.

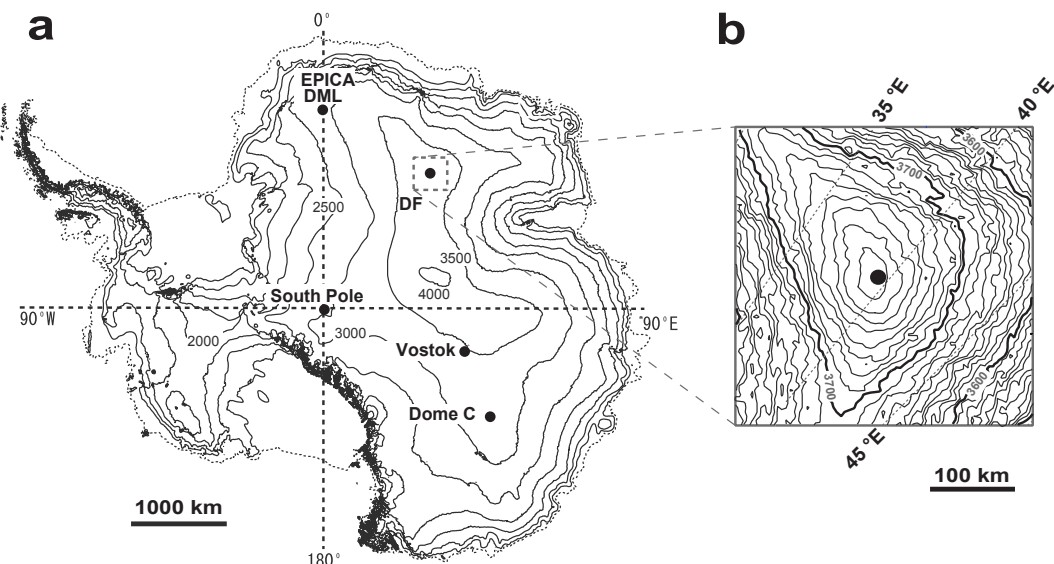

**Figure 1.** Maps of (a) the whole of Antarctica and (b) the area around Dome Fuji. Surface elevation values (in metres) are based on the digital elevation model of Bamber et al. (2009).

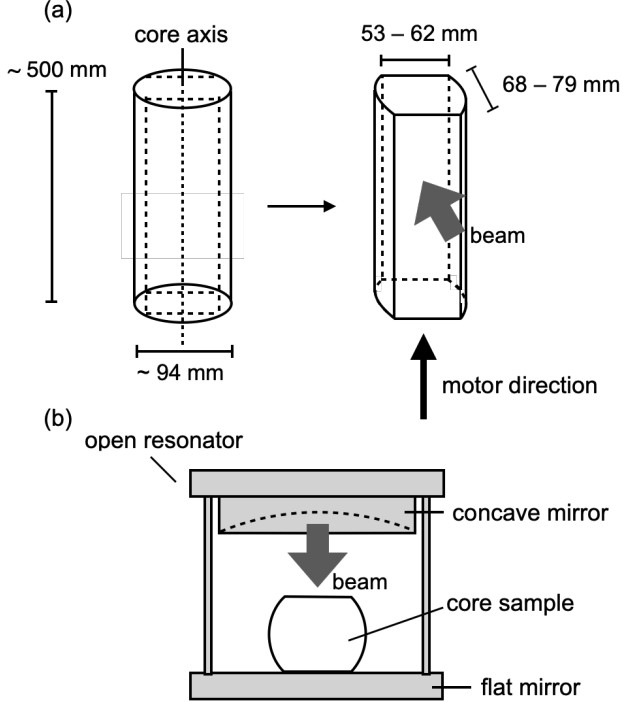

**Figure 2.** Diagrams of the (a) core cutting geometry and (b) experimental setup (viewed from the front).

## 3. Results

### 3.1 Continuous variations in $\Delta\varepsilon$

Figure 3 presents typical examples of the continuous variation of $\Delta\varepsilon$ along 0.5 m core samples, based on ice core samples acquired at depths of 125.0–125.5, 1300.0–1300.5 and 2398.5–2399.0 m. The span of the y-axis equals the dielectric anisotropy of a single ice crystal (0.0334) at –30 °C (see Saruya et al., 2022). Small fluctuations of $\Delta\varepsilon$ over distances from 0.02 to 0.5 m are apparent, indicating minor but significant variations in the COF within each 0.5 m long piece of ice core. The mean values and standard deviations for each 0.02 m long portion were derived based on different TEM $_{0, 0, q}$ resonance modes using the open resonator method. In the case of the example data presented in Fig. 3, the mean values (standard deviations) were 0.0076 (0.0005), 0.0194 (0.0006) and 0.0300 (0.0005) for the 125.0, 1300.0 and 2398.5 m depth samples, respectively.

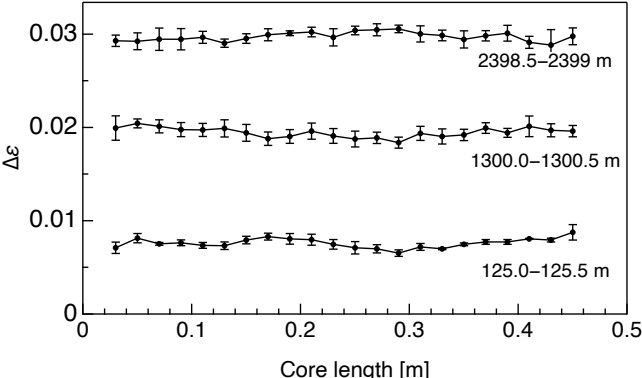

**Figure 3.** Examples of variations in $\Delta\varepsilon$ along 0.5 m long ice core samples obtained from continuous measurements. The span of the y-axis equals the dielectric anisotropy of a single ice crystal. Error bars indicate standard deviations obtained from different resonance modes of the open resonator. Typically, eight different modes were obtained from each single 0.02 m portion.

### 3.2 Depth-dependent variation in $\Delta\varepsilon$

Figure 4 shows the variations in the mean $\Delta\varepsilon$ values (Fig. 4a), the standard deviations (S.D.s) of $\Delta\varepsilon$ values (Fig. 4b), the detrended $\Delta\varepsilon$ values (defined as the difference between each data point and the third-order polynomial fitting curve to the $\Delta\varepsilon$ data; Fig. 4c), and the oxygen isotope ratio values ($\delta^{18}$O; Fig. 4d). The latter data were obtained from Dome Fuji Ice Core

Project Members (2017). The mean values and S.D.s were determined at intervals of approximately 0.5 m along the core sample, using approximately 23 data points for each interval. The detrended $\Delta\varepsilon$ data represent the relative degree of $c$-axis clustering and the extent of deformation relative to those in the surrounding depth. These detrended values are more useful than the original values as a means of assessing fluctuations in the COF and for comparison with other physicochemical properties.

Figure 4a demonstrates that the overall trend of the $\Delta\varepsilon$ values increase with increasing depth, although small fluctuations are evident within a length scale of approximately 100 m. These variations in $\Delta\varepsilon$ also appear to be continuous rather than abrupt. Large decreases in $\Delta\varepsilon$ are also apparent at depths of 1800, 2150 and 2300 m as indicated by the three arrows in Fig. 4a. The depths of 1800 and 2300 m correspond to the transition periods from glacial to interglacial, while the depth of 2150 m corresponds to Marine Isotope Stage (MIS) 7a, b, c. At shallower depths above approximately 300 m the $\Delta\varepsilon$ values are instead relatively constant and the detrended $\Delta\varepsilon$ values are larger than the general mean of the data.

Figure 4b indicates that the S.D. values for each approximately 0.5 m long core sample exhibit both a long-term trend and short-term fluctuations. Over the whole dataset, the S.D.s are approximately constant down to a depth of about 1300 m but then increase between that point and 1800 m. During the termination II event, the S.D. values exhibit a rapid decrease while, below 2000 m, large increases appear at approximately 2150 and 2300 m, and a decrease in S.D. is also seen during the termination III event. The variations in the detrended $\Delta\varepsilon$ values in Fig. 4c reflect the fluctuations noted above. In this panel, the grey shading indicates three periods, from the early stage of each interglacial period to the termination event. In this plot, positive/negative values indicate a high/low degree of $c$-axis clustering relative to the regression line in terms of depth, as a consequence of specific mechanisms. The amplitude of these fluctuations becomes larger at greater depth. As an example, the amplitude at shallow depths (< 500 m) is approximately 0.001, while that at greater depths (> 1800 m) is 0.003.

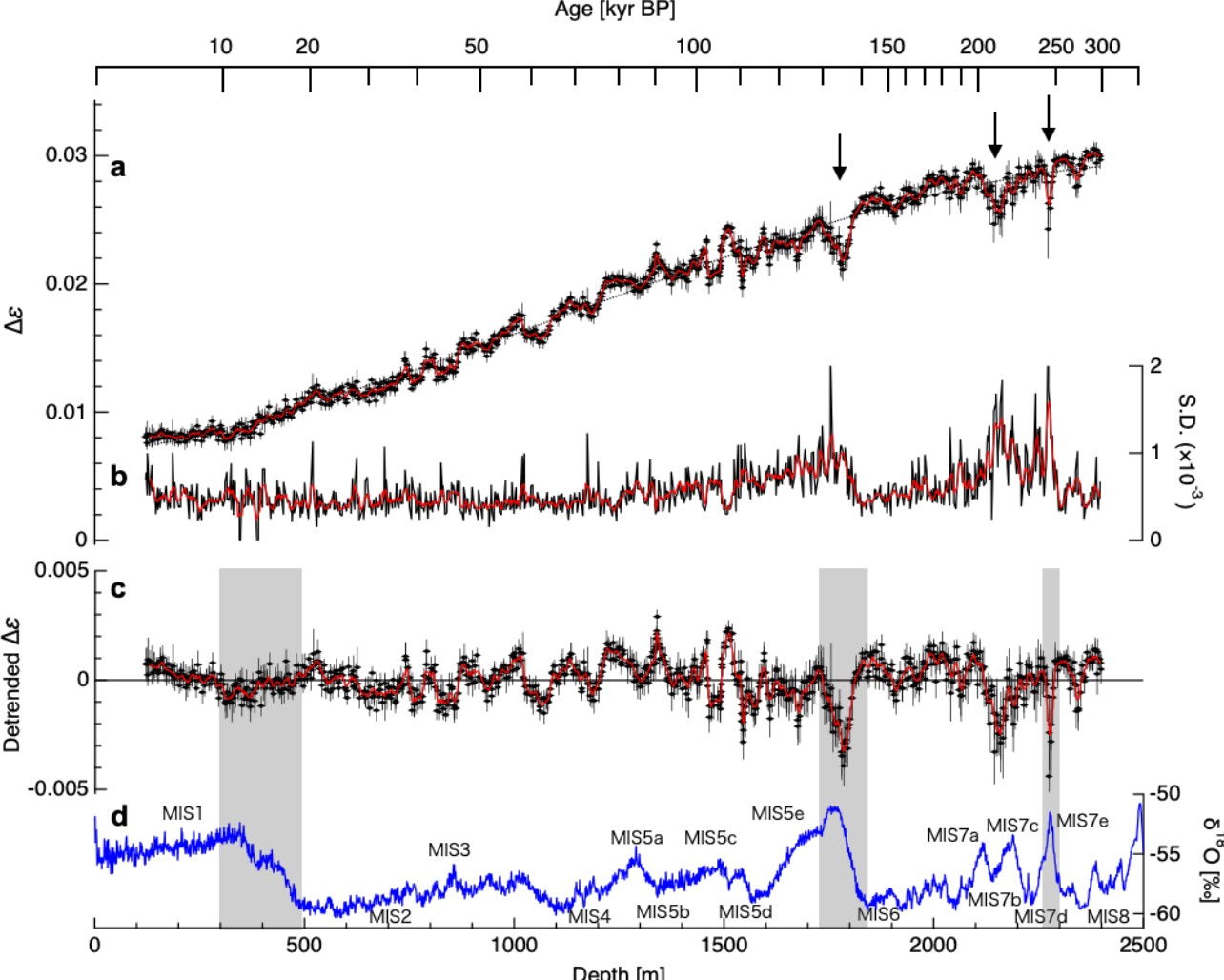

**Figure 4.** (a) Variations in $\Delta\varepsilon$ along the DF1 ice core depth, showing the mean values for each core section (approximately 0.5 m in length). Error bars correspond to the standard deviation (S.D.) within each core while the dotted line indicates a third-order fitting and the three arrows indicate depths associated with significant decreases. (b) The depth profile of the S.D. values for $\Delta\varepsilon$. (c) Detrended $\Delta\varepsilon$ values, defined as deviations from the third-order fitting to the data in panel (a). The red lines in plots (a–c) were generated by smoothing at 10 m intervals. (d) Oxygen isotope ratios ($\delta^{18}O$) in the DF1 ice core (modified from Dome Fuji Ice Core Members, 2017). The grey bands indicate the transition periods from glacial to interglacial. Marine Isotope Stage (MIS) events are also shown.

### 3.3 Eigenvalues derived from $\Delta\varepsilon$

In previous studies, COF development was examined based on variations in the normalized eigenvalues $a_1^{(2)}$, $a_2^{(2)}$ and $a_3^{(2)}$. To allow a direct comparison with other ice cores, we therefore derived normalized eigenvalues from the present $\Delta\varepsilon$ data.

The magnitude of $a_3^{(2)}$ indicates the extent of clustering of the *c*-axes toward the vertical that is the same as the core axis. In fact, the value of $a_3^{(2)}$ has been shown to increase with increasing depth in ice cores drilled at dome summits due to *c*-axis

clustering. Saruya et al. (2022) reported that the relationship between $\Delta\varepsilon$ and these eigenvalues is:

$$\Delta\varepsilon = \Delta\varepsilon_s \, (a_3^{(2)} - (a_1^{(2)} + a_2^{(2)}) / 2). \qquad (1)$$

Here, $\Delta\varepsilon_s$ is the dielectric anisotropy of a single ice crystal. In the case of the present measurements at –30 °C, $\Delta\varepsilon_s$ was determined to be 0.0334 (see the appendix in Saruya et al., 2022). Using the relationship $a_1^{(2)} + a_2^{(2)} + a_3^{(2)} = 1$ and assuming that $a_1^{(2)}$ and $a_2^{(2)}$ are approximately equal (that is, horizontal isotropy), equation (1) can be rewritten as:

$$a_3^{(2)} = (2\Delta\varepsilon/\Delta\varepsilon_s + 1) / 3. \qquad (2)$$

Using these equations, we were able to derive the eigenvalues from the $\Delta\varepsilon$ data. Assuming that the approximation noted above is valid, the normalized eigenvalues obtained from earlier COF studies could then be directly compared with $\Delta\varepsilon$ data derived using our newly developed method that we present here. Figure 5 provides such a comparison between eigenvalues estimated from $\Delta\varepsilon$ and those generated using an optical method on the basis of the DF1 core samples (modified from Azuma

et al., 2000). Here, the black and red lines indicate dielectrically derived (that is, thick-section-based) and optically derived (that is, thin-section-based) values, respectively. In Fig. 5a, we compare the thick-section-based $\Delta\varepsilon$ and thin-section-based eigenvalue anisotropy values defined as $\Delta a^{(2)} = a_3^{(2)} - (a_1^{(2)} + a_2^{(2)})/2$. Figure 5b compares the normalized eigenvalue components obtained from the thick-section-based and thin-section-based approaches. In both panels, the fluctuations of the thick-section-based eigenvalues and the anisotropy are smaller than those of the corresponding thin-section-based values.

Since the thin-section-based eigenvalues were determined using sections with thicknesses of approximately 0.5 mm, the normalized eigenvalues reflect the statistically averaged *c*-axis clustering of several hundred to thousands of ice grains. In contrast, a single thick-section-based $\Delta\varepsilon$ data point is representative of an ice specimen as thick as 33–79 mm. Therefore, the sampling volumes between the two methods differ by a factor of 85–190. In addition, the thick-section-based eigenvalues presented here are the averaged values for each 0.5 m long core, meaning that the sampling volumes actually differ by more

than three orders of magnitude. Thin-section measurements can provide information regarding localized distributions of *c*-axis orientations for each grain within thin-sections, while thick-section measurements indicate the COF characteristics on a bulk scale. An obvious difference between the thick-section- and thin-section-based eigenvalues is the size of the fluctuations and the continuity of the data distribution. Specifically, the thin-section-based eigenvalues exhibit sudden fluctuations well above 0.1 within many depth ranges that are not observed in the thick-section-based eigenvalues. Figure 6

presents a modified version of the comparison in Fig. 5a based on a direct comparison between $\Delta\varepsilon$ values for each 0.02 m interval and thin-section-based eigenvalue anisotropy data. Even using the raw $\Delta\varepsilon$ data without averaging over each 0.5 m long ice core, the scatter of the thin-section-based eigenvalue anisotropy values is typically far greater.

Figure 7 plots the variations in the eigenvalues for the horizontal direction derived from the thick-section-based measurements (that is, $a_1^{(2)}$ or $a_2^{(2)}$ as shown in Fig. 5b). The right axis indicates the permittivity values corresponding to the normalized eigenvalues on the left axis.

Saruya et al. (2022) reported the relationship $\varepsilon_x = \varepsilon_\perp + \Delta\varepsilon_s\, a_1^{(2)}$, where $\varepsilon_x$ is the relative permittivity along the principal x axis and $\varepsilon_\perp$ is the permittivity perpendicular to the $c$-axis. Thus, if $\varepsilon_x$ and $\varepsilon_y$ (the relative permittivity along the principal y axis) are approximately equal, these permittivity components will vary over the range of $\Delta\varepsilon_s\, a_1^{(2)}$. The eigenvalue $a_1^{(2)}$ changes from 0 to 1 while $\Delta\varepsilon_s = 0.0334$. A value for $\varepsilon_\perp$ of 3.1367 is also provided in the appendix to Saruya et al. (2022). Using this relationship, we can compare eigenvalues and permittivities.

The size and fluctuation of the horizontal eigenvalue is directly related to the magnitude of the permittivity, and thus to the refractive index or speed of radio waves within the ice sheet. In addition, the fluctuation size and the change in fluctuation with depth provide reliable information concerning the magnitude of ice-fabric-based radio wave reflections within the ice sheet. Specifically, $\varepsilon$ changes as small as 0.002 (the typical magnitude of changes below approximately 1500 m) are sufficient to cause internal radio echo reflections (with strengths of approximately –75 dB) that are detectable by ice radar instrumentation (e.g., Fig. 1 in Fujita et al., 1994 and Fig. 10 in Fujita et al., 2000). Thus, the large depressions of $\Delta\varepsilon$ as well as the small-scale fluctuations in $\Delta\varepsilon$ should be detectable using ice sounding radar (Fujita et al., 1999).

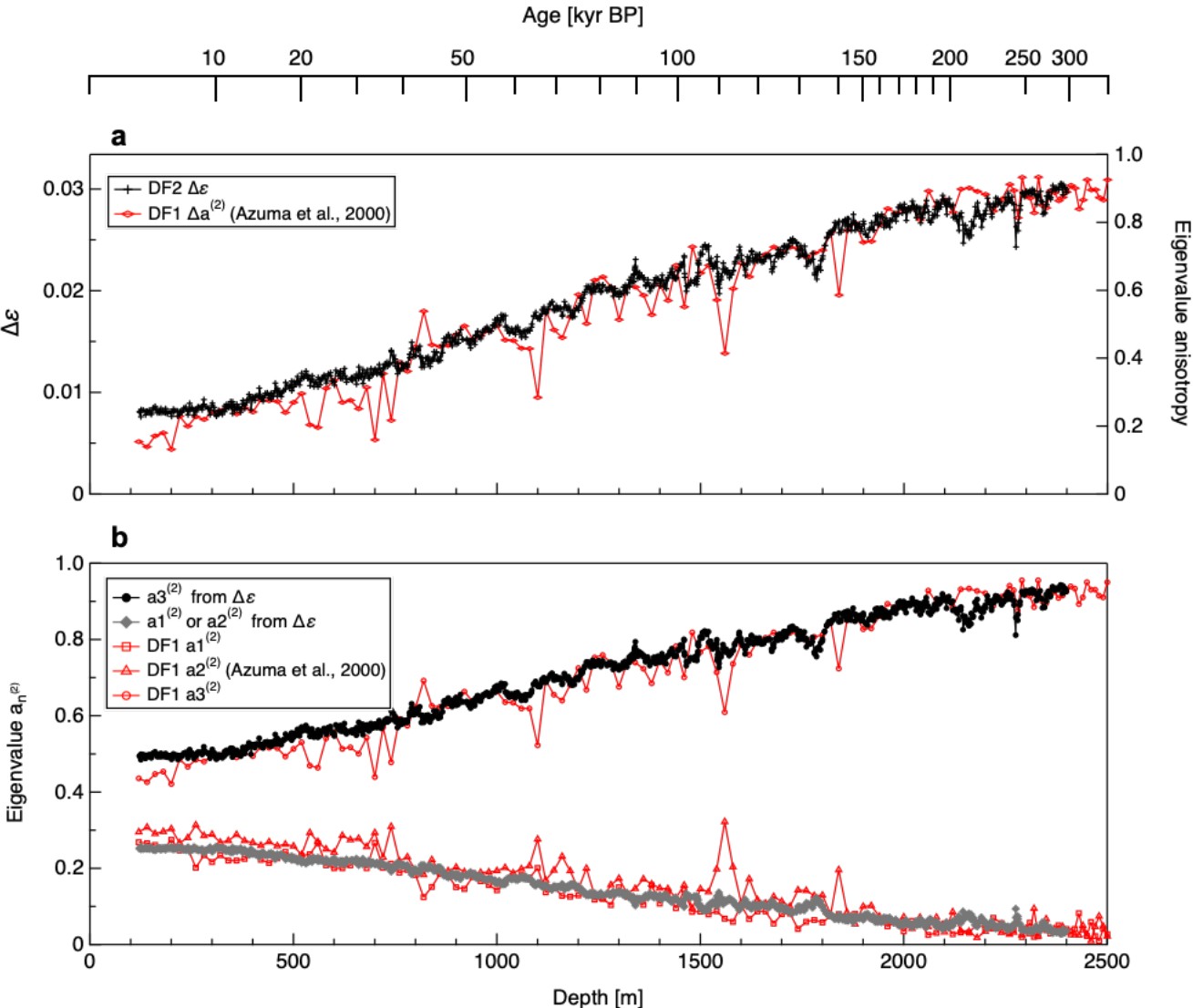

**Figure 5.** Comparison of eigenvalues obtained from the present dielectric measurements and from thin-section measurements. (a) The thick-section-based $\Delta\varepsilon$ and optically derived (thin-section-based) eigenvalue anisotropy data for DF1, (b) a comparison of thick-section and thin-section-based normalized eigenvalue components. Black/grey and red lines indicate thick-section-based and thin-section-based values, respectively. The DF1 eigenvalues are taken from Azuma et al. (2000).

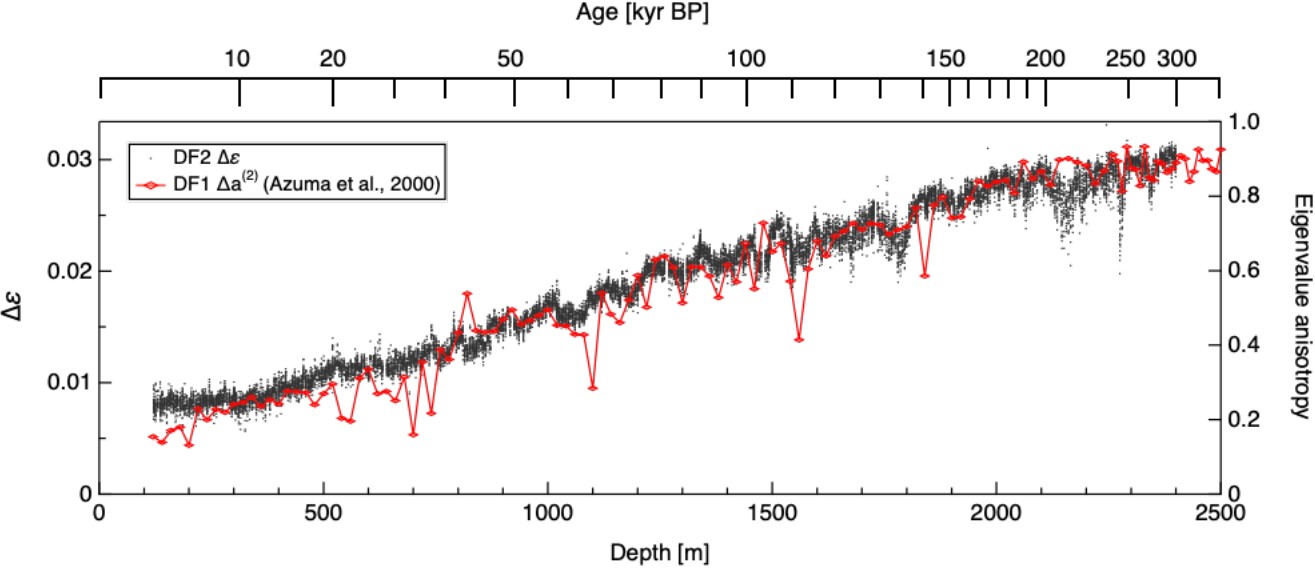

**Figure 6.** A modified version of the data in Fig. 5a. The raw $\Delta\varepsilon$ values at each 0.02 m step are shown instead of means over each 0.5 m. The scatter of the thin-section-based data (red symbols and lines) is far larger than that of the thick-section-based data.

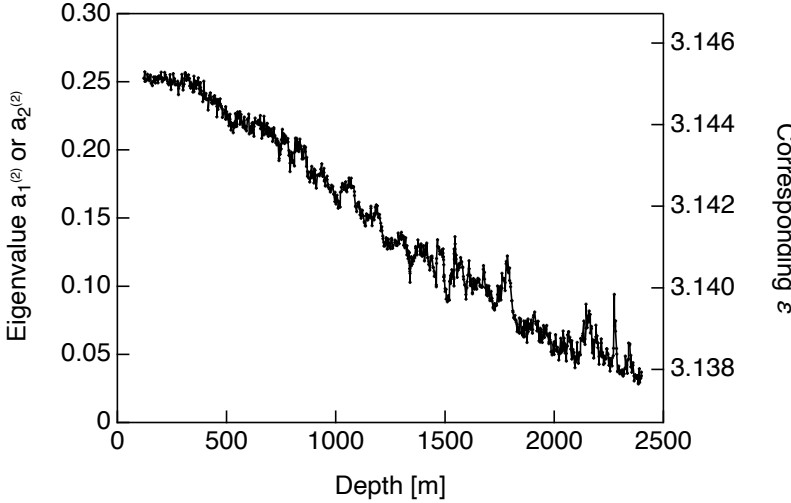

**Figure 7.** Eigenvalues in the horizontal direction derived from dielectric measurements (that is, $a_1^{(2)}$ or $a_2^{(2)}$ in Fig. 5b). The right-side y-axis is the permittivity corresponding to the eigenvalue on the left-side y-axis. See Saruya et al. (2022) for the relationship between the normalized eigenvalues and permittivity.

## 3.4 Comparison of the present data with DF1 and EDC ice cores

Figure 8 summarizes the development of the normalized eigenvalues $a_3^{(2)}$ along the DF2, DF1 and EDC ice cores. Here, we use the normalized eigenvalues instead of $\Delta\varepsilon$ values, and the magnitude of $a_3^{(2)}$ reflects the degree of $c$-axis clustering toward the core axis, just as $\Delta\varepsilon$ does. The DF1 and EDC data are from Azuma et al. (2000) and Durand et al. (2009), respectively, and are derived from thin-section measurements. Note that the glaciological conditions in DF and EDC are similar. The ice thickness, annual accumulation rate and mean surface temperature values for DF are $3028 \pm 15$ m, $27.3 \pm 1.5$ kg m$^{-2}$ year$^{-1}$

and $-54.4$ °C (Dome Fuji Ice Core Project Members, 2017) while those for EDC are $3309 \pm 22$ m, $25 \pm 1.5$ kg m$^{-2}$ year$^{-1}$ and $-54.5$ °C (EPICA Community Members, 2004). The conversion from depth to age was performed using Supplemental Materials in Bazin et al. (2013) and Dome Fuji Ice Core Members (2017). The general data trend is the same for both cores, in that the $a_3^{(2)}$ values increase with increasing depth. However, the small fluctuations over spans of less than 10 kyrs are different. Durand et al. (2007) reported a rapid increase in $c$-axis clustering along with a decrease in grain size during the

termination II event (that is, the transition from interglacial to glacial) in the EDC ice core. In contrast, such variations were not observed in the case of the termination I event. This difference was attributed to a transition to enhanced horizontal shear in the glacial ice in conjunction with the termination II event, although our own view is different. Durand et al. (2007) did not report a change in association with the termination III event, but a possible strengthening of $c$-axis clustering does appear at this point. Durand et al. (2007) also suggested that a 60 m thick layer indicating reduced clustering of the $c$-axes exists

below 1680 m (approximately 125 kyrs BP) and corresponds to the MIS5e event. In Durand's data, the local minimum in the degree of $c$-axis clustering was accompanied by a local maximum in the deuterium concentration. It should also be noted that the variation trends observed at the termination-II/MIS5e and termination-III/MIS7e events in the EDC core were approximately the same as those in our measurements.

      Although glaciological conditions (such as surface temperature, accumulation rate and ice thickness) are similar at the EDC

and DF, the development of COF within the DF1 ice core (as determined using thin sections) is inconsistent with those in the EDC core and with our own analysis of DF2 core samples. As stated in Section 3.3, the limited statistical reliability of the thin-section-based method prevents a reliable comparison.

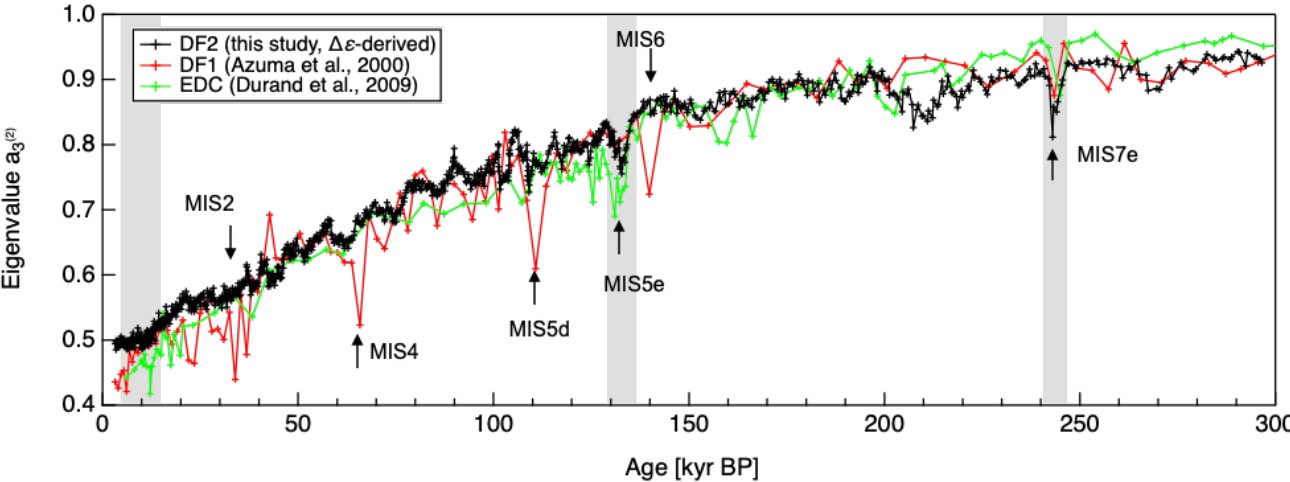

**Figure 8.** Comparisons of $a_3^{(2)}$ eigenvalues between the DF2 ice core (thick-section-based data), DF1 ice core (thin-section-based data) and EDC ice core (thin-section-based). The DF1 and EDC data are from Azuma et al. (2000) and Durand et al. (2009), respectively. Arrows indicate MIS events noted by Durand et al. (2007). Grey shading indicates interglacial to glacial transitions. A common age scale referred to as DFO2006 was applied (Dome Fuji Ice Core Project Members, 2017).

## 4. Discussion

### 4.1 General trend in the variation of $\Delta\varepsilon$

#### 4.1.1 Basic facts and questions

As a basis for the present discussion, we first need to determine if the observed variations in $\Delta\varepsilon$ are significant or simply the result of measurement error. Saruya et al. (2022) reported that such errors were minimized by solving equations for multiple resonance frequencies simultaneously to find a unique solution for $\varepsilon$. They determined the total error in $\varepsilon$ to be $-0.01 \pm 0.01$, and this systematic error was primarily attributed to the limited widths of the ice core samples. The data show dielectric anisotropy in the horizontal direction (that is, perpendicular to the core axis) in addition to the vertical direction, which is a potential source of error when determining the depth-dependent variation of $\Delta\varepsilon$ (Saruya et al., 2022). The COF in the DF ice core exhibits so-called single pole fabric characteristics. However, as a result of an imbalance in the strain in the horizontal directions, this single pole fabric shows elliptically elongated distributions (Azuma et al., 1999, 2000; Saruya et al., 2022). Saruya et al. (2022) reported that the error in $\Delta\varepsilon$ could be as large as 10–15% in extreme cases based on accidental core rotation occurring in conjunction with irregular core breaks at the drilling site. According to Saruya et al. (2022), accidental core rotation is a rare event that can occur a few or several times within every 1000 m length of the core. In addition, the probability of the maximum error (that is, 10–15% of $\Delta\varepsilon$) is small. In the case of accidental (that is, abrupt) rotation of the ice

core, the mean value of the error will be half the maximum. Thus, the data must be examined to identify any suspiciously abrupt steps/jumps in $\Delta\varepsilon$. One such inspection within the brittle zone between 600 and 900 m identified suspicious abrupt

changes in $\Delta\varepsilon$ values at depths of 750 and 800 m. Because the ice core samples in this zone are sometimes brittle, the continuity of the core in terms of core orientation could have been broken. These abrupt changes in $\Delta\varepsilon$ could have therefore resulted from accidental core rotation. In contrast, the $\Delta\varepsilon$ values at other depths were found to change continuously, without any anomalous steps/jumps. On this basis, with the exception of the brittle zone, we believe that the evident variations in $\Delta\varepsilon$ are true reflections of continuous changes in COF development, and that the present data contain only minor systematic

errors in $\Delta\varepsilon$, the magnitude of which changes at several depths. We also note that the S.D. values for the $\Delta\varepsilon$ data were only minimally affected by possible rotations of the core. Saruya et al. (2022) have shown that changes in cluster strength occur simultaneously in all horizontal orientations.

Complex permittivity data obtained for ice based on analyses using MHz frequencies were reviewed by Fujita et al. (2000). The real part of the complex permittivity of the ice in ice sheets is a function of the COF as well as the density, the

315 concentration of soluble impurities (primarily acidic impurities) and the temperature. In contrast, both the hydrostatic pressure and the shape of air bubbles have relatively minor effects. In addition, the effect of plastic deformation can be significant and needs to be investigated further. What we are interested in here is whether or not some factors other than COF can modify the anisotropic values of permittivity. In the case of ice containing bubbles, the density, soluble impurity concentration and temperature do not modify the dielectric anisotropy (Fujita et al., 2000). Thus, COF is the only factor

responsible for anisotropic permittivity in the polycrystalline ice in the ice sheet. In addition, to date, grain boundaries, dust inclusions, clathrate hydrate inclusions or salt inclusions in ice have not been shown to produce detectable changes in permittivity.

It is also important to note that the $\Delta\varepsilon$ values were fully compatible with the normalized COF eigenvalues assuming a single pole fabric with $c$-axis clustering along the vertical direction (Saruya et al., 2022). Therefore, the degree of clustering can be

expressed using $\Delta\varepsilon$ instead of the normalized COF eigenvalues for the sake of simplicity. The overall trend of the $\Delta\varepsilon$ values was to increase down to a depth of 2400 m (Fig. 4a). This trend was consistent with previous findings that the degree of $c$-axis clustering is strengthened at greater depths within the dome summits of ice sheets, as described in the Introduction (Sect. 1) to this paper. This large-scale trend is explained primarily by the rotation of the $c$-axes toward the compressional axis associated with dislocation creep. The data also indicate continuous variations within depth scales on the order of 10 to $10^2$

330 m. In particular, the three depressions indicated by arrows in Fig. 4a at depths corresponding to interglacial/glacial transitions at approximately 1800 and 2300 m and to the MIS7abc event at 2150 m are significant. These results raise many questions and it would be helpful to identify the following: (i) the factors controlling variations associated with changes in time and depth, either initial microstructural conditions, effects of impurities that modify dislocation movements and/or microstructure, positive/negative feedback effects from COF evolutions, or a complex mixture/interplay of these, (ii) the

335 reasons for the increased fluctuating amplitude of $\Delta\varepsilon$ over depth scales on the order of 10 to $10^2$ m with increasing depth, (iii) the reasons for the increase in the S.D. of $\Delta\varepsilon$ values with increasing depth, (iv) the further growth of these variations under

shear and at deeper englacial environments, and (v) means of applying our new understanding of DF to wider range of ice sheets. Answering these questions may lead us to a better understanding of ice rheology.

### 4.1.2 Correlation between $\Delta\varepsilon$ and its standard deviation

Figure 9 shows a comparison between detrended $\Delta\varepsilon$ values and the associated S.D.s with smoothing over 10 m intervals. Note that the y-axis in this plot is inverted to make visual inspection easier. A striking feature is that, below about 1200 m, the inverted detrended $\Delta\varepsilon$ data are well correlated with the S.D. values. The linear correlation coefficient below 1200 m is –0.75 while that at depths shallower than 1200 m is –0.09. Because the detrended $\Delta\varepsilon$ represents the relative degree of $c$-axis clustering and the extent of deformation relative to the surrounding depth, this high degree of correlation means that

more/less deformed ice had smaller/larger fluctuations within the ice core sample. These small- and large-scale variations are likely to be related and, in the following section, we investigate the cause of variations in the cluster strength of $c$-axes.

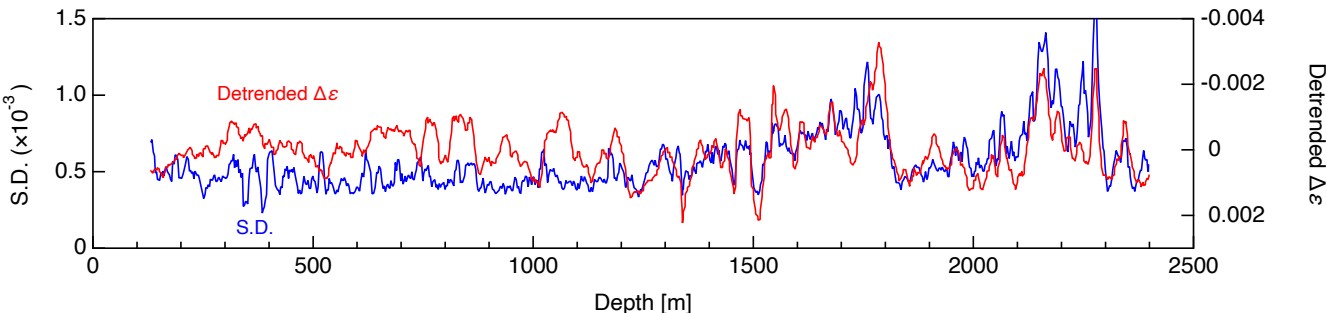

**Figure 9.** Comparison of the S.D. values of $\Delta\varepsilon$ data determined at intervals of approximately 0.5 m along the core sample, using

approximately 23 data points for each interval (blue line) and detrended $\Delta\varepsilon$ data defined as deviations from the third-order-fitting to the data in Fig. 4a (red lines) (smoothed over 10 m intervals). Note that the y-axis for the detrended $\Delta\varepsilon$ has been inverted.

### 4.2 Comparison of $\Delta\varepsilon$ with physicochemical properties in the DF ice core

To address the questions raised in Section 4.1, we attempted to establish correlations between the detrended $\Delta\varepsilon$ values and

impurities and crystal grain size data acquired for the DF1 ice core. These factors could possibly affect the behaviour of deformation (e.g., Paterson, 1991; Cuffey et al., 2000; Cuffey and Paterson, 2010; Fujita et al., 2014, 2016; Saruya et al., 2019). It is known that glacial ice includes many soluble impurities as well as dust particles. Furthermore, glacial ice exhibits finer grains and more rapid deformation (that is, a large value of the flow-enhancement factor) in comparison with interglacial ice (e.g., Paterson, 1991). However, the direct influence of these factors on COF development is unclear.

### 4.2.1 Relationship to soluble impurities and dust particles

Figure 10 plots the detrended $\Delta\varepsilon$ data, the corresponding S.D. values, $\delta^{18}O$, grain size, concentrations of soluble impurities ($Cl^-$, $SO_4^{2-}$ and $Ca^{2+}$), concentrations of HCl and NaCl, concentrations of sulphate salts and concentration of dust particles along the DF1 ice core. Note that the soluble impurities, HCl, NaCl, sulphate salt, dust particle and grain size data are taken from the literature (Goto-Azuma et al., 2019; Iizuka et al., 2012; Dome Fuji Ice Core Project Members, 2017; Azuma et al., 2000, respectively). As stated in Section 3.2, significant decreases in the $\Delta\varepsilon$ values are apparent at approximately 130, 210 and 245 kyrs BP, during which the detrended $\Delta\varepsilon$ values can be as low as –0.0035. The periods corresponding to these large depressions (along with another period around 10 kyrs BP) are indicated by grey shading. Within these time spans, we can observe correlations between the large depressions in the detrended $\Delta\varepsilon$ plot and lower concentrations of soluble impurities, larger grain sizes and low concentrations of dust. These regions correspond to the termination-I/MIS1 (AI), termination-II/MIS5e (AII), termination-III/MIS7e (AIII) and MIS7abc (A7) transition periods. We hereinafter refer to this type of relationship as Type A. Note that the labels in Fig. 10 indicate the termination or MIS stage in each case. In addition to these large depressions, smaller decreases in the detrended $\Delta\varepsilon$ values that occur in conjunction with changes in the concentrations of impurities at depths below 130 kyrs BP are also apparent. Within these periods, we can observe higher concentrations of soluble impurities, smaller grain sizes and higher concentrations of dust, as indicated by the brown shading. This type of relationship is defined herein as Type B (numbered in order according to increasing depth in Fig. 10). Although periods with higher concentrations of soluble impurities and dust particles are apparent at 15–35 and 60–70 kyrs BP (that is, at shallower depths), clear depressions in the detrended $\Delta\varepsilon$ values (as seen at greater depths) do not appear. Thus, obvious correlations between detrended $\Delta\varepsilon$ values and these factors are primarily restricted to greater depths. In the following section, we discuss the causes of the variations in $\Delta\varepsilon$ in both the Type A and B relationships, focusing on the influence of soluble impurities and dust particles.

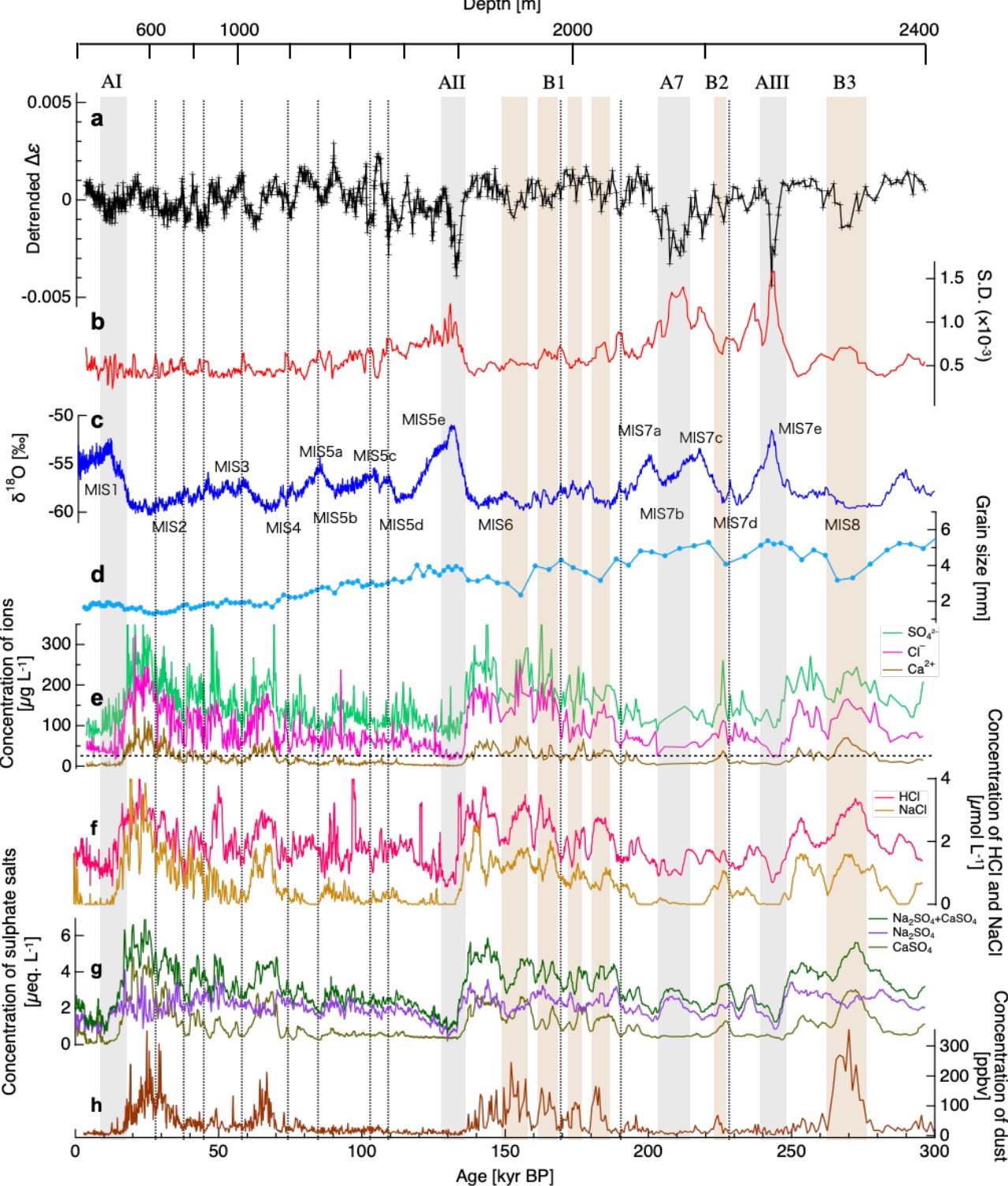

**Figure 10.** Comparison of detrended $\Delta\varepsilon$ values with other data from the DF1 ice core. Plots of (a) detrended $\Delta\varepsilon$, (b) S.D. of $\Delta\varepsilon$ with 10 m smoothing, (c) $\delta^{18}O$ (Dome Fuji Ice Core Project Members, 2017), (d) grain size (Azuma et al., 2000), (e) concentrations of $Cl^-$, $SO_4^{2-}$ and $Ca^{2+}$ ions (Goto-Azuma et al., 2019), (f) concentrations of HCl and NaCl, (g) concentration of sulphate salts (~5 m smoothing) (Iizuka et al., 2012), and (h) concentration of dust particles (Dome Fuji Ice Core Project Members, 2017) as functions of ice age. Grey shading indicates correlations between the large depressions in the detrended $\Delta\varepsilon$ values and lower concentrations of soluble impurities (defined as Type A). These regions correspond to the termination-I/MIS1 (AI), termination-II/MIS5e (AII) and termination-III/MIS7e (AIII) transition periods and to MIS7abc (A7). Note that the labels indicate the termination or MIS stage in each case. Brown shading indicates correlations between the depressions in detrended $\Delta\varepsilon$ values and higher concentrations of dust (defined as Type B, numbered in order according to increasing depth).

The vertical dotted lines in the plots indicate small increases in the S.D. values corresponding to Antarctic isotope maxima in terms of $\delta^{18}O$ and decreased levels of $Cl^-$. The horizontal dashed line in plot (e) indicates the minimum $Cl^-$ concentration.

## 4.2.2 Effect of chloride ions

Various soluble impurities, including $Cl^-$, $SO_4^{2-}$ and $Ca^{2+}$ ions, have been examined in terms of their deformation enhancement effect (e.g., Nakamura and Jones, 1970; Hörhold et al., 2012; Freitag et al., 2013; Hammonds and Baker, 2018). However, the depth-dependent variations of $Cl^-$, $SO_4^{2-}$ and $Ca^{2+}$ ions are similar; therefore it is difficult to identify the most important ion species from the time-dependent profiles. The correlation coefficients between the detrended $\Delta\varepsilon$ values and the $Cl^-$, $SO_4^{2-}$ and $Ca^{2+}$ concentrations were determined to be 0.21, 0.16 and 0.21, respectively (as estimated from data extracted at 5 m intervals between 130 and 2400 m, $n = 455$). However, to the best of our knowledge, only $Cl^-$, $F^-$ and $NH_4^+$ ions have been shown to modify the dislocation movement (including the dislocation density) within the ice crystal lattice when substituted for $H_2O$ molecules (Jones, 1967; Jones and Glen, 1969; Nakamura and Jones, 1970). Fujita et al. (2014, 2016) hypothesized that layered deformation in firn results from a combination of the texture initially formed by seasonal variations in metamorphism and the effects of ions such as $Cl^-$, $F^-$ and $NH_4^+$. The same group also attributed high correlations between the concentration of $Ca^{2+}$ ions and deformation (reported by Hörhold et al., 2012; Freitag et al., 2013) to seasonal synchronization with cycles of $Cl^-$, $F^-$ and $NH_4^+$ ions and seasonal variations in metamorphism. Although data regarding $Cl^-$, $SO_4^{2-}$ and $Ca^{2+}$ ions are included in Fig. 10e, we suggest that only $Cl^-$ ions have the effect of softening the ice, while $SO_4^{2-}$ and $Ca^{2+}$ do not play any direct role in terms of substitution for $H_2O$ molecules. Typically, the concentration of $Cl^-$ ions is much higher than those of $F^-$ and $NH_4^+$ ions in Antarctic ice cores (e.g., Udisti et al., 2004), and so the present study focuses on the concentration of $Cl^-$ ions. Dissolved and substituted $Cl^-$ ions can increase the dislocation (point defect) density in ice and promote dislocation movement, which in turn will result in active plastic deformation and $c$-axes clustering. Therefore, the Type A relationship could be explained by variations in the level of $Cl^-$ ions in the ice. It should also be noted that the distribution of $Cl^-$ ions in firn and ice is readily homogenized by various diffusion mechanisms taking place in the solid, liquid or vapour phase (e.g., Barnes et al., 2003). In such cases, rather than the development of layered, heterogeneous deformation, the $Cl^-$ ions would be expected to promote the homogeneous deformation of the firn and ice

(Fujita et al., 2016). This effect explains the aspect of the Type A relationship in which limited homogenization of the Cl⁻ ion distribution causes inhomogeneous layer deformation. In warm periods, the S.D. values associated with the $\Delta\varepsilon$ might be expected to increase because the extent of homogeneous deformation is restricted as a consequence of the low concentration of Cl⁻ ions. We further note that the S.D.s of the $\Delta\varepsilon$ values indicating the degree of fluctuation within each ice core sample (that is, over distances of less than 0.5 m) increased not only in association with the Type A relationships but also at many of the Antarctic isotope maxima (AIM) events during glacial periods. Local decreases in the Cl⁻ ion concentration (indicated by the vertical dotted lines in Fig. 10) are also evident, suggesting that the level of Cl⁻ ions played an important role in determining the amount of deformation and degree of homogeneity.

Within the Type A regions, the depressions in the detrended $\Delta\varepsilon$ on the glacial side (that is, the older side) can be explained by rapid decreases in the concentration of Cl⁻ ions. However, the depression at the interglacial side (the younger side) cannot be attributed to the same cause because the concentration of Cl⁻ ions is slightly increased going toward the interglacial period. Therefore, the cause of the rapid development of COF during interglacial periods remains unclear. Considering the effects of Cl⁻ ions on dislocation movements, the amount of HCl is more important than that of NaCl. HCl can dissolve in ice while dissociating to release Cl⁻ ions, while NaCl will exist as solid particles. Therefore, the concentration of HCl is considered to be directly correlated with the concentration of discharged Cl⁻ ions. The variations in the HCl and NaCl concentrations over time are shown in Fig. 10f. Watanabe et al. (2003) and Iizuka et al. (2012) reported the Cl⁻ and NaCl concentrations in the DF ice core over the most recent 300 kyrs, respectively, and we were able to derive HCl concentrations from the differences between the Cl⁻ and NaCl concentrations. The correlation coefficients for the relationships between the detrended $\Delta\varepsilon$ values and the HCl and NaCl concentrations were found to be 0.31 and 0.11, respectively (as estimated from data extracted at 5 m intervals between 130 and 2400 m, $n = 455$), showing a weak correlation only between the detrended $\Delta\varepsilon$ and HCl concentration data. This result implies that the release of Cl⁻ ions by HCl is an important factor influencing the dislocation movement and development of COF.

### 4.2.3 Effect of dust particles

The effect of dispersed particles on ice deformation has been investigated by various laboratory experiments, although conflicting results are reported with either softening or hardening of the ice (Cuffey and Paterson, 2010). From the present data, it is apparent that the decreases in the detrended $\Delta\varepsilon$ data at the regions associated with Type B relationships are associated with higher dust concentrations. On the basis of this relationship, we suggest that dust particles tend to impede COF clustering. The concentrations of Cl⁻ ions were also found to be high at the Type B locations but, even in such situations, it appears that c-axis clustering was restricted by the presence of dust particles. Consequently, we propose that the relative strength of COF clustering is mainly determined by a balance between the levels of Cl⁻ ions and dust particles. If the effects of Cl⁻ ions (which include promoting dislocation movement and increasing deformation) are stronger than the effect of dust particles (which limits c-axis clustering), the degree of c-axis clustering could be enhanced. However, in the B1–B3

locations, the degree of *c*-axes clustering was found to be less than in adjacent layers even though the Cl⁻ ion concentrations were quite high. Therefore, the reduced *c*-axes clustering brought about by the dust particles was evidently more powerful than the deformation enhancement resulting from the Cl⁻ ions.

We suggest two possible reasons for reduced *c*-axis clustering: (i) restricted deformation due to the dislocation inhibition effect of dust particles and (ii) the various mechanisms that contribute to deformation (other than dislocation creep).

In the first case, if the deformation of an ice sheet proceeds solely via dislocation creep, weak *c*-axis clustering indicates that the degree of deformation must be impeded by dust particles. The hardening of artificial polycrystalline ice following the addition of high concentrations of sand particles was reported by Hooke et al. (1972), who suggested that sand particles surrounded by tangled networks of dislocations impeded dislocation movement. This effect could restrict both deformation and *c*-axis clustering. In the second case, deformation mechanisms other than dislocation creep could contribute to deformation, with smaller crystal grains in Type B relationships being a potential cause of reduced COF clustering. In one example, Azuma et al. (2000) proposed that the weakening of *c*-axis clustering is caused by the contribution of diffusional creep that does not contribute to the *c*-axis rotation. According to Azuma's group, the contribution of diffusional creep at depths having finer grains significantly increases in the DF ice core. The ice sheet conditions (that is, the pressure and temperature) in Antarctica are situated within a boundary zone between dislocation and diffusional creep on the deformation mechanism map (e.g., Shoji and Higashi, 1978; Goodman et al., 1981; Duval et al., 1983). Therefore, the contribution of diffusional creep might be significant at depths with smaller grains. In this case, a weakening of *c*-axis clustering does not necessarily indicate a restriction of the extent of deformation. In this study, we are not able to resolve the possible effects of grain size and the presence/absence of diffusional creep. Although periods with higher concentrations of dust particles are evident around 25 and 65 kyrs BP, these regions are not associated with decreased $\Delta\varepsilon$ values. Because the extent of deformation is minimal at shallower depths, it is likely that the effect of dust particles was not yet significant at these locations.

### 4.2.4 Influence of salt particles

Salt particles could also possibly affect COF development and are known to exist in polar ice cores at volume fractions much larger than those of dust particles (Ohno et al., 2005). However, the amount of salt particles is not reflected in the dust profile in DF1 (Fig. 10h). The time-based profiles of the sulphate salt ($Na_2SO_4$ and $CaSO_4$) concentration data obtained from Iizuka et al. (2012) are shown in Fig. 10g. Although the concentrations of salt particles in the DF1 ice core were not determined, Iizuka et al. (2012) estimated sulphate salt concentrations using the relationship between ion balance and the chemical compounds found in salt inclusions (Iizuka et al., 2008). The resulting plots of salt concentrations over time are similar to the profiles of the Cl⁻ ion and dust particle concentrations. Salt particles might be expected to act as solid particles and so could impede *c*-axis clustering. However, the formation of salt particles is associated with the generation of HCl, which is likely to activate dislocation movement (e.g., Iizuka et al., 2012; Fujita et al., 2016). In addition, sulphate acids can become salt

particles by reacting with dust particles (Ohno et al., 2006). If the salt particles both impede and enhance deformation, their contribution to the degree of $c$-axis clustering could be determined by the balance between these effects. The influence of salt particles was not considered in previous studies; however, it might be important to both deformation and COF development.

### 4.3 Growth of variation amplitude in $\Delta\varepsilon$

The growth of the variation amplitude associated with the $\Delta\varepsilon$ fluctuations provides insights into the nature of the deformation process. In the case of the Type A relationships, the depressions in $\Delta\varepsilon$ are small within the AI region but deeper at AII, A7 and AIII. These results demonstrate that contrasting rheology was preserved all the way to deeper layers, so that the extent of clustering was weak compared with the surrounding layers. An initial shear strain would be expected to promote further deformation of the COF, because the ice would be softer due to a positive feedback mechanism (Azuma, 1994). At very

shallow depths, at which the COF is almost random, the $c$-axes start to rotate toward the compressional axis as deformation progresses. In this early stage, the rotation of these $c$-axes in this manner increases the density of slip planes close to the plane of maximum shear stress (45° from the compressional axis). This deformation enhancement is temporarily higher than that of the initial random fabric, such that the initial deformation softens the ice in a positive feedback loop. In contrast, as the fabric continues to develop, the ice becomes harder as these $c$-axes rotate closer to the compressional axis. In the case of

the majority of the crystal grains, the slip planes will tend to rotate away from the plane of maximum shear stress and so the ice will become progressively harder in a negative feedback loop. However, we note that the present rheology contrasts were not caused by positive feedback of the rheology due to the COF. In the case of vertical compression such as occurs at DF, the COF-based enhancement factor increases slightly during the very initial stage of deformation, after which the enhancement factor monotonically decreases (Azuma, 1994). The positive detrended $\Delta\varepsilon$ values indicating enhanced $c$-axis

clustering are attributed to restrictions of dislocation movement with increased deformation due to the work hardening resulting from dislocation pile-up. Therefore, excessive deformation and $c$-axis clustering is limited even in layers with high levels of $Cl^-$ ions. In contrast to the Type A relationships, the depressions associated with Type B relationships are minimal regardless of the depth or age, suggesting the absence of feedback mechanisms in the case in which the DF is subjected primarily to vertical compression.

### 4.4 Initial conditions in microstructures

    At the point of bubble close-off, where the transition from firn to ice occurs (approximately 100 m in depth), $\Delta\varepsilon$ is already about 0.008, which is approximately 25% of the value for a single ice crystal (Saruya et al., 2022). In fact, $\Delta\varepsilon$ values of this magnitude have also been observed at the base of the firn (that is, at the top of the bubble-containing ice; Fujita et al., 2009, 2014, 2016). At this depth in DF, there is almost no contribution of the dielectric polarization effect due to the vertical

elongation of pore spaces and the ice matrix (Fujita et al., 2009). X-ray diffraction analyses of the DF firn have also

demonstrated that the COF $c$-axes tend to cluster around the vertical direction or become inclined near this bubble close-off horizon depending on the sample (Fujita et al., 2009). More recently, similar results showing a stacked layer COF pattern were reported in snow within a 2 m deep pit at a plateau site in East Antarctica (Calonne et al., 2017). Going from the top to the bottom of the firn, a sequence exhibiting typical deformation phenomena was observed (Fujita et al., 2016), with

515 variations in density, impurity concentration and dielectric properties as well as in the correlations between these parameters (see Table 7 in Fujita et al., 2016). It is therefore likely that $\Delta\varepsilon$ at a depth of approximately 100 m represents a superposition of the initial COF caused through metamorphism at the near-surface depth and subsequent metamorphism and deformation of the firn. This initial phenomenon is likely to be greatly affected by the presence of Cl⁻ ions and dust particles because vertical deformation of the ice is dominant in firn. Because Cl⁻ ions have the greatest effect on the densification of firn

(Fujita et al., 2016), we suggest that these ions are among the main factors determining the depth of the transition from firn to bubble-containing ice, known as the lock-in depth (LID). It is also likely that the $\Delta\varepsilon$ values along deep ice cores are valuable indicators that can be used to determine the timescale of the LID in detail. In addition, these $\Delta\varepsilon$ values will be directly correlated with the vertical thinning of each layer and so can be extremely useful as a means of refining ice core dating models and providing constraints on strain values. Because both the LID and vertical thinning are related to the strain

rate enhancement caused by the COF, the presence of Cl⁻ ions and the temperature of the ice, it is apparent that the variations in the LID over time and the cumulative vertical thinning in each layer should be closely related. This possibility should be examined in future studies.

## 4.5 Implications for the deformation regime in ice sheets

The evolution of the COF clustering strength was investigated herein based on variations in $\Delta\varepsilon$. On this basis, we suggest

that the five questions posed in Section 4.1 can be answered as follows.

(i) The factors determining the time- and depth-dependent variations of COF clustering are the initial microstructural conditions that occur at near-surface depths, the degree of deformation within the firn, the levels of ionic impurities that promote dislocation movement throughout the deformation processes and the amount of dust particles that tend to impede clustering. Because vertical compression is a major component of deformation, the positive and negative feedback effects of

535 deformation enhancement associated with COF evolution will not play a major role other than to provide weak positive feedback during the very initial stage of deformation and weak negative feedback in the later stage of deformation (Azuma, 1994).

(ii) The large decreases in $\Delta\varepsilon$ seen within specific depth ranges are attributed to Cl⁻ ions and dust particles, both of which primarily increase the amplitude of the $\Delta\varepsilon$ fluctuations within distances on the order of 10 to $10^2$ m with increasing depth.

Considering the effects of Cl⁻ ions in terms of promoting dislocation movement, the amount of HCl is more important to COF development than that of NaCl. Many, if not all, of the $\Delta\varepsilon$ fluctuations below 1200 m can be explained by these effects. However, we also observed fluctuations that cannot be explained by the concentrations of Cl⁻ ions or dust particles alone. In

particular, the causes of COF fluctuation at shallower depths (corresponding to the glacial period between AI and AII, see Fig. 10) are still unclear, and so further investigation of other factors determining COF development is required. $F^-$ and $NH_4^+$

ions are potential additional candidates. Although we have no data concerning the $F^-$ and $NH_4^+$ concentrations within the DF ice core, these ions have been shown to modify dislocation movement in ice in laboratory experiments and in polar ice sheets. The effects of salt particles on COF development should also be clarified. Salt particles could potentially act as solid particles to impede $c$-axis clustering, while the formation of salt particles is closely associated with the generation of HCl that can promote dislocation movement. Because the volume fraction of salt particles is much larger than that of dust

particles, the former would be expected to have a greater effect on microstructural evolution and deformation.

(iii) It is also highly likely that the same factors listed above were responsible for the increased S.D. values at greater depths. It should additionally be noted that, in the case of low $Cl^-$ ion concentrations, there were more significant increases in S.D.

(iv) The present work examined the COF within the upper 80% of the DF ice core. It is highly probable that the dome position migrated in the past (see Sect. 2.1). Thus, the current profiles of layered COF fluctuations will have direct

implications regarding further deformation. For example, we would expect to encounter various stress/strain configurations resulting from conditions near the base, such as ice flow, undulating bedrock topography and/or ice-thickness-dependent partial melting (Dome Fuji Ice Core Project Members, 2017). Under such variable conditions, in addition to the two major factors of $Cl^-$ ions and dust particles, the layered COF fluctuations will have large effects in terms of enhancing or impeding deformation. The less clustered COF will be either softer or harder because it contains a greater variety of crystal orientation

(and thus slip planes of hexagonal ice). Azuma and Goto-Azuma (1996) discussed the deformation of ice sheets with perturbed and layered clustering of single pole COF and suggested the occurrence of heterogeneous layered thinning leading to layer folding or boudinage. In fact, layering disturbances and folding at the lower regions of ice cores have been previously observed on several occasions (e.g., Svensson et al., 2005; Faria et al., 2010; Jansen et al., 2016). As an example, Jansen et al. (2016) investigated small-scale disturbances at the bottom part of the NEEM ice core based on numerical

modelling and concluded that the folding structures were initiated by the formation of bands in which the lattice was tilted relative to the bulk COF. This conclusion is in agreement with a prior report by Azuma and Goto-Azuma (1996). Although visual inspections have not identified layering disturbances in the EDC ice core, Durand et al. (2009) found significant fluctuations in eigenvalues. This same group established that there were no clear correlations between the fabric fluctuations and climate or chemical composition. Below a depth of 2846 m, very small grains (less than 1 mm in size) appeared between

larger grains, indicating the onset of crystal nucleation. Although signs of migration recrystallization, such as interlocking grain boundaries, were not observed, their findings suggest the possibility of migration recrystallization at sufficiently high temperatures. Thus, further and more detailed investigations are required for a better understanding of the COF development and deformation regime in the deeper parts of the core. Currently, the retrieval of continuous ice core records corresponding to ages of more than 1 Myrs is an important challenge in palaeoclimatology (see topic of this special issue). Identifying

suitable sites for the drilling of very old ice will require knowledge of the subglacial topography and englacial layering. Radar sounding is a powerful means of observing englacial layering and can evidently detect enhanced COF layering at

deeper layers (see Fig. 7 in this paper and Fujita et al., 1999, 2000). When identifying candidate sites using ice sounding radar, it will be important to distinguish between stable layering and heterogeneous thickness layers. Specifically, the presence of layers with heterogeneous thickness could indicate initiation of anomalous strain and thus layer disturbances.

(v) Finally, we suggest an important implication obtained from this study. Layered sequences of ice core signals resulting from Cl⁻ and dust particles are commonly obtained from inland ice core sites over the entire widths of the Antarctic ice sheet with minor local variations. Because both Cl⁻ and dust particles are primary factors controlling the development of COF (as demonstrated by this study), the COF layering profiles established by these factors should be similar in many ice core sites located in inland of the Antarctic ice sheet. Profiles of COF layering should be similar not only at ice core sites but also over much wider areas of the ice sheet having common sequences of both Cl⁻ and dust particles. Indeed, in Figure 8, the variations in $a_3^{(2)}$ eigenvalues over time in the DF2 ice core (this study) and the EDC ice core are seen to be similar even though these two sites are approximately 2000 km apart in East Antarctica. This is the first example of common features of COF variations within two very remote ice cores and is thus an important finding. Because radar data can be used to detect COF layering, it should be possible to compare deep COF layers across very wide areas of ice sheets, and such analyses should be performed at other locations.

## 5 Conclusion

With the aim of obtaining a better understanding of the deformation regime in ice sheets, we assessed the viability of using dielectric anisotropy, $\Delta\varepsilon$, as a new indicator of crystal orientation fabric (COF) using ice core samples taken from Dome Fuji in East Antarctica. This method is a useful means of determining the degree of COF vertical clustering resulting from vertical compressional strain at the dome. The present investigation covered the upper 80% of the entire dome thickness, from depths of 100 to 2400 m, representing an ice cover to an age of approximately 300 kyrs BP. Examining thick, 1 m long ice core specimens acquired at 5 m intervals, this study was able to generate high-resolution COF data. Compared with existing thin-section-based methods, the new method described herein provided information with greatly improved statistical significance. The major findings of the present study can be summarized as follows.

- The data establish that the overall trend of the $\Delta\varepsilon$ values was to increase with increasing depth and also show that $\Delta\varepsilon$ fluctuated over distance scales in the range of 10–10² m. This general pattern in which the values increase is consistent with previous findings that the *c*-axes of ice crystals concentrate toward the core axis due to grain rotation caused by uniaxial compression. In addition, we discovered large depressions in $\Delta\varepsilon$ during three major transition periods from glacial to interglacial (termination-I/MIS1, termination-II/MIS5e, termination-III/MIS7e) as well as the MIS7abc event. These results indicate that deformation variations occurred in a continuous manner from the near-surface to deeper layers. Moreover, fluctuations in $\Delta\varepsilon$ over distances of less than 0.5 m, as reflected by S.D. values, were inversely

correlated with $\Delta\varepsilon$ at depths greater than 1200 m, meaning that such fluctuations were enhanced during the glacial/interglacial transition periods.

• A positive correlation between $\Delta\varepsilon$ and the concentration of Cl$^-$ ions along with a negative correlation with the concentration of dust particles in the ice core were also established in those regions associated with significant decreases in $\Delta\varepsilon$. Based on these results, we propose that there are several factors that may potentially affect COF clustering with changes in time and depth. These include the initial COF that is formed by metamorphism at near-surface depths as well as ionic impurities, such as Cl$^-$ ions, and dust particles. Cl$^-$ ions released from HCl are known to

increase the dislocation density and to promote dislocation movement throughout the deformation process, while dense concentrations of dust particles may impede COF clustering. An additional factor is the difference between the amounts of Cl$^-$ ions and dust particles. These parameters mainly determine the amplitude of the variations in $\Delta\varepsilon$ over distances on the order of 10 to $10^2$ m and are also responsible for the increase in the S.D. of the $\Delta\varepsilon$ values with increasing depth.

• The present data also have important implications concerning the deformation/flow of ice sheets, as discussed in Sect.

4.5(iv). We suggest that the COF structure (and thus the deformation structure) of polar ice sheets should be evaluated by focusing on the presence of impurities, the density of dust particles and COF layering, as well as changes in these factors. Importantly, the present study demonstrated small perturbations of COF clustering in the ice sheet, showing growth of the COF contrast amplitude at deeper layers. In addition, it should be emphasized that layered sequences of ice core signals related to Cl$^-$ and dust particles were common at inland ice core sites within the wide Antarctic ice

sheet. Because both Cl$^-$ and dust particles are among the major factors determining the development of COF, profiles of COF layering established by these factors toward very deep depths should be similar within many ice core sites located in the inland regions of the Antarctic ice sheet and even in wider areas of the ice sheet having similar sequences of both Cl$^-$ and dust particles.

• It should also be noted that VHF/UHF radar sounding is a useful technique that provides information concerning

permittivity contrast (and thus COF contrast) within the deep interior of polar ice sheets, and such analyses should be examined further in the future. Importantly, when searching for sites suitable for obtaining core samples of very old ice, we must be careful to avoid layers with heterogeneous thicknesses within the lowest approximately 20% of the ice sheet, as determined by ice sounding radar, because the presence of heterogeneous thicknesses indicates the initiation of disturbances in layered structures due to effective horizontal strains.

*Data availability*

The dielectric anisotropy data will be published in the National Institute of Polar Research ADS data repository in conjunction with the publication of the present manuscript in The Cryosphere.

*Author contributions*

We list author contributions using a standard called CrediT (Ghan et al., 2016) to achieve greater clarity in contributions of all authors. TS: Conceptualization, Methodology, Validation, Formal analysis, Investigation, Data curation, Writing - Original draft, Visualization. SF: Conceptualization, Methodology, Validation, Formal analysis, Investigation, Writing - Original draft, Supervision, Project administration, Funding acquisition. YI, AM, HO, AH and WS: Writing - Review & editing. MH and KG-A: Methodology, Validation, Investigation, Writing - Review & editing.

*Competing interests*

The authors declare that they have no conflict of interests.

*Acknowledgements*

We thank the handling editor Kaitlin Keegan and two anonymous reviewers for helpful comments and suggestions. The authors are grateful to all the Dome Fuji Deep Ice Core Project members who contributed to obtaining the ice core samples, 650 either through logistics, drilling or core processing. The main logistics support was provided by the Japanese Antarctic Research Expedition (JARE), managed by the Ministry of Education, Culture, Sports, Science and Technology (MEXT). This work was supported by JSPS KAKENHI Grant Number 18H05294. We thank Gaël Durand for providing eigenvalue data from the EDC ice core.

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
