# Peer review of "Development of crystal orientation fabric in the Dome Fuji ice core in East Antarctica: implications for the deformation regime in ice sheets"

_The Cryosphere, 2021_

## Author Comment (AC1)

Reply to the reviewer 1

We thank the reviewer for careful review of our manuscript and thoughtful comments to improve it. In the following, we describe our responses (in blue) point-by-point to each of reviewer's comment (in black).

Dear editor,

This paper investigates the crystal orientation fabric of the Dome Fuji Station ice core using a novel methodology as the dielectric anisotropy (from the dielectric permittivity tensor). Dielectric anisotropy is revealed as a good indicator of the vertical clustering of the crystal orientation fabric, also exhibiting a correlation with the concentration of chloride ions and with the amount of dust particles in the ice core. From the results, the authors conclude that the COF clustering is therefore affected by the presence of chloride irons, which increase the dislocation density, promote dislocation creep and enhance the COF clustering, while the presence of dust impedes it. The results show a COF layering in the upper 80% of the ice sheet, where the COF contrast amplitude increases at deeper layers. The conclusion is sound regarding the lowest 20% of the ice sheet, as layers will behave differently under stress depending on the COF cluster strength.

This well-written and well-organised manuscript presents a very useful methodology to be further evaluated in the future. We could obtain the COF contrast from the permittivity contrast obtained in VHF/UHF radar sounding. This will allow comparing deep COF layers avoiding areas with layers with heterogeneous thickness. This heterogeneity can lead to layer disturbances and folding due simple shear close to the bedrock.

The detailed description of the presence of soluble impurities and particles and its correlation with the COF is very useful for the understanding on the effect of them on ice rheology, which is currently unknown.

The manuscript is relevant for The Cryosphere and thus can be a valuable contribution once some important issues are addressed. Thus, I recommend that the paper is accepted after minor revisions.

We thank the reviewer a lot for the summary above.

Suggestions for improvement:

Line 50: this statement "It has been suggested that the finer grain size in glacial ice results from high concentrations of impurities such as dust particles or soluble substances that restrict grain growth via pinning and drag at the grain boundaries" requires a reference.

We will add references, for example,

Alley, R.B., and Woods, G.A.: Impurity influence on normal grain growth in the GISP2 ice core, Greenland, J. Glaciol., 42(141), 255-260, https://doi.org/10.3189/S0022143000004111, 1996.

Gow, A.J., Meese, D.A., Alley, R.B., Fitzpatrick, J.J., Anandakrishnan, S., Woods, G.A., and Elder, B.C.: Physical and structural properties of the Greenland Ice Sheet Project 2 ice core: A review, J. Geophys. Res., 102(C12): 26559-26575, https://doi.org/10.1029/97JC00165, 1997.

Line 80: I would give details of the physicochemical properties obtained.

We did not fully understand what the reviewer intended to comment. If it was meant so that the authors should give details of the physicochemical properties obtained, at this stage of the introduction, our idea to revise this part as follows.

Present sentence:
*The resulting data were compared with various physicochemical properties obtained from analyses of the DF ice core to better understand the factors influencing COF development.*

Idea of revision:
*The resulting data were compared with various chemical and physical properties of ice, such as major ions, amount of dusts, salt inclusions, grain size and so on, obtained from analyses of the DF ice core to better understand the factors influencing COF development.*

Table 1: Could you explain in the text why the thickness at the EDC samples (Durand) are not indicated?

In two papers by Durand et al., we did not find information on the thickness at the EDC samples. However, because they used thin-section-based method, we guess it was approximately 0.5 mm or slightly less. We will explain it in the main text too.

Line 110: the authors do not explain why this study focuses on the upper 80% and not in the whole ice core. What are the difficulties to apply it in the bottom 20%? A discussion on this aspect will be useful.

To make readers better understanding, we will add following sentences:
*It is noted that we expect that the interpretation of the COF from dielectric measurements will be*

*challenging below 2400 m due to the inclined layers and extremely coarse crystal grains. The layered structures began to be inclined relative to the horizontal layer; the inclination is less than 5º above 2400 m, however, it reaches 20º at 2800 m, and 50º at 3000 m (Dome Fuji Ice Core Project Members, 2017). Additionally, the extremely huge coarse grains (with grain size of > 50 cm) are observed at the deepest part from visual inspection. The influences of these factors should be confirmed by experiments in the future. Therefore, we conducted the measurements to a depth of 2400 m in present paper.*

Figure 2. Colouring in red and blue the lines is not necessary as they do not provide extra information.

We will modify it as suggested.

Line 150: the detrended Ae value is a key parameter used in this work. It is mathematically defined in the text, but it would be very useful for readers to be able to see an explanation of what it value means, in practical terms.

We will add a following sentence at around Line 154:
*The detrended $\Delta\varepsilon$ represents the relative degree of c-axis clustering and the extent of deformation relative to the surrounding depth.*

Line 186. has the value of 0.0334 for the single ice crystal been determined in this study or does it come from the literature? In this case, a citation would be needed.

Thank you for pointing it out. Indeed, it comes from Appendix in Saruya et al., (2022). We will add "Appendix in Saruya et al., (2022)". (This paper was finally published in 2022.)

Line 213. Would it be possible to briefly explain the relationship between the permittivity value and the normalized eigenvalues? (here or in the caption of figure 6). I find the reference to Saruya et al. 2021 not enough, as this data is relevant for the conclusions.

We will add following sentences in the main text.
*In Saruya et al., (2022), it was explained as,*
$$\varepsilon_x = \varepsilon_\perp + \Delta\varepsilon\, a_1^{(2)}$$
*in equation 5 (where $\varepsilon_x$ is the relative permittivity along the principal axis x, $\varepsilon_\perp$ is the permittivity perpendicular to the c-axis). That is, if $\varepsilon_x$ and $\varepsilon_y$ (the relative permittivity along the principal axis y)*

*are approximately equal, these permittivity components have variable range as $\Delta\varepsilon\,a_1^{(2)}$.*

*Eigenvalues $a_1^{(2)}$ change from 0 to 1, while $\Delta\varepsilon$ =0.0334. $\varepsilon_\perp$ is also given in appendix in Saruya et al., (2022), as 3.1367. Using this relation, we can compare the values of eigenvalues and permittivity.*

We will add explanation about it in the revised manuscript.

Figures 4 and 5: I suggest including the references at the legend, as in figure 7.

We will add a reference at the legend. Reference for the DF1 core is Azuma et al., (2000).

In general: please, check the graphics in all figures. Box and axis markers do not match (as in figure 7).

In figure 7, we slightly separated the bottommost tick from lower axis for the visibility. (Standoff function in Igor Pro). It was made in purpose.

Line 384: the reason why the presence of HCl has a stronger effect on dislocation migration than NaCl is explained later, in line 387. I suggest moving line 384 there to make the paragraph more understandable.

We will move the sentence.

Line 444: In general: It should be explained with a bit more detail, what the positive or negative feedback mechanism referred to Azuma (1994) does mean (Relationship between CPO and deformation conditions).

We will add sentences meaning following contents:
*At very shallow depth where COF is close to random, the c-axes start to rotate toward compressional axis as the deformation progress. At this beginning stage, c-axes' rotation toward compressional axis cause temporally increase population (density) of slip planes close to the plane of maximum shear stress (45 degrees from the compressional axis). Then, deformation enhancement is temporally higher than that of the initial random fabric. This means, initial deformation made the ice softer; it is positive feedback. In contrast, as the development of fabric progresses more, ice becomes harder since these c-axes rotate further closer to the compressional axis; for majority population of the crystal grains, slip planes tend to rotate away from the plane of maximum shear stress. Then ice will be harder progressively; it is negative*

*feedback.*

Line 499: Regarding the alteration of layers in the deep parts in ice sheets, here I miss some discussion with observations already done in ice cores (as in Faria et al., 2010; Jansen et al., 2016, etc…).

Thank you for your suggestion. We will add discussion about layer disturbance at the bottom part as follows:
*Actually, layering disturbances and folding at the bottom part of ice cores have been observed in various ice cores (e.g., Svensson et al., 2005; Faria et al., 2010; Jansen et al., 2016). For example, Jansen et al. (2016) investigated the small-scale disturbances at bottom parts in the NEEM ice core from numerical modeling and concluded that the folding structures were initiated by the formation of tilted-lattice bands relative to the bulk COF. This conclusion seems to agree with the suggestions by Azuma and Goto-Azuma (1996).*
*In the EDC ice core, although the layering disturbances are not reported from visual inspections, highly fluctuated development of eigenvalues has been observed in Durand et al. (2009). Moreover, they reported there were no clear correlation between the fabric fluctuations and climate or chemical compositions. Below a depth of 2846 m, very small grains less than 1 mm were appeared between the large grains, which means the starts of crystal nucleation. Although the evidences of migration recrystallization such as interlocking grain boundaries were not observed, they suggest the possibility of migration recrystallization as the temperature is high enough. Thus, further and detailed investigations are required for a better understanding of the COF development and deformation regime at deeper part.*

Conclusion and chapter 4.5 Implications for the deformation regime in ice sheets: both texts are very similar. I would modify the conclusion part in bullet points or in a more synthetised way, because as it is now it reads as a repetition of the explanation given in the previous section 4.5.

We agree with your suggestion. We will modify and simplify the "Conclusion" section with bullet.

---

## Author Comment (AC2)

Reply to the reviewer 2

We thank the reviewer for careful review of our manuscript and thoughtful comments to improve it. In the following, we describe our responses (in blue) point-by-point to each of reviewer's comment (in black).

General comments

This contribution provides an excellent methodology for exploring the chemical and mechanical heterogeneity of ice, with a likelihood of inferring crystallographic fabric from permittivity anisotropy. The approach is valuable for the community and the data appear robust. I have no concerns about the data acquisition. The comparisons with the nearby cores and with the Dome Fuji 1 core chemistry make good sense, including using the orientation tensor as a metric. This is a large dataset that will serve a purpose for many years to come.

I have a few significant concerns about the interpretations. Some of these can be addressed with additional explanation, and some may require reevaluating the text.

We thank the reviewer for the appreciation and summary above.

Specific comments

1a. Crystal orientation fabric (COF) is not the only factor that affects permittivity or permittivity anisotropy. Dust, salts, or other impurities that are layered in the ice core, even at a fine scale, can cause permittivity anisotropy. I suggest that the paper review the potential impact of these factors on anisotropy and evaluate whether they can robustly related the permittivity data to COF.

We will add following sentences in "4.1.1 Basic facts and questions" (Discussion section):

*Data on the complex permittivity of ice around megahertz frequencies are reviewed by Fujita et al. (2000). The real part of the complex permittivity of ice in the ice sheets is a function of several controlling factors as follows: (1) COF, (2) density, (3) impurity concentration (mainly acidity), and (4) temperature. In contrast, both (5) hydrostatic pressure and (6) air-bubble shape have relatively minor effects. The effect of (7) plastic deformation can be significant and needs to be investigated further. We explain in more details. In ice with bubbles, either density, impurity or temperature has no effects on the dielectric anisotropy. There has been no data that can raise a possibility that grain boundaries, dust inclusions, clathrate hydrate inclusions or salt inclusions within ice can have detectable impact on the permittivity. Matsuoka et al. (1997, 1998) investigated the influence of soluble impurities on dielectric properties by measuring the permittivity of impurities doped ice. They reported that the small amount of impurities did not significantly affect dielectric properties.*

1b. If this investigation cannot rule out impurities as factors, then I suggest that the interpretations, including the discussion and conclusion, focus more on reporting the permittivity anisotropy and its correlation with the other features in Fig. 9 and less on COF. I recognize that several sections in the discussion consider how the impurities affect COF, all of which appear to be valid and substantive ideas. At the same time, the lack of a consistent relationship between permittivity anisotropy and, e.g., Cl and dust, indicates that the mechanisms are quite incompletely understood. I do not feel that the data and reasoning support the interpretation (line 400) "Consequently, we propose that the relative strength of COF clustering is mainly determined by a balance between the levels of Cl- ions and dust particles."

As answered in #1a, we can rule out chloride ions or dust particles from influenceable factors to the permittivity values.

2. I was not able to understand the data collection methods from the text, in particular the geometry of the sampling. A figure that shows the spatial relationship between the core, the samples, and the measurement and motor directions would be extremely useful.

Thank you for your suggestions. Experimental procedures and diagrams are detailed in Saruya et al. (2022, this paper was finally published in 2022); however, we will add experimental diagrams in revised manuscript to make readers better understanding. An example is shown in below:

[Figure]

Fig.# Schematic diagrams of (a) the core cutting and (b) experimental setup (view from the front side).

3a. The text does not include an explanation of the source of uncertainty. It appears that the reported standard deviation is the result of some form of averaging, and it is not clear whether any systematic uncertainty is factored it. I suggest the manuscript add a clear method for calculating uncertainty.

We will add following sentences in "4.1.1 Basic facts and questions" (Discussion section):
*Estimation of errors are detailed in Saruya et al. (2022). They reported that errors were minimized by solving equations for multiple resonance frequencies simultaneously to find a unique solution for $\varepsilon$. The final errors in $\varepsilon$ were –0.01 ± 0.01. The systematic error is mainly caused by limited widths of the ice core sample.*

3b. On the topic of choosing which technique to use to analyze a core, lines 209-210 state that the "statistical validity of the thin-section-based method is inferior to that of the thick- section-based method." I don't find that statement accurate. The thick section data unquestionably average over a larger volume, but that doesn't mean that they are more statistically valid. I do think that representing the larger volume will provide a better relationship to rheology than the potentially high-frequency variations recorded in thin section data, but that is not the claim currently made. Additionally, as implied by my comment #1, the relationship between COF and permittivity anisotropy is not necessarily straightforward.

We will remove "statistical validity of the thin-section-based method is inferior to that of the thick-section-based method" because this sentence is not suitable for an explanation of thin-section measurements. However, we consider that an increase in sample volume produces statistical significance because thick-section results are comparable with superimposed results of more than 100 thin-sections. In revised manuscript, we will explain the advantages of thin-section measurement and thick-section measurement as follows:
*Thin-section measurements can provide local features with distributions of c-axis orientations in each grain, while thick-section measurements can provide bulk and representative features of the COF.*
Regarding the relationship between COF and dielectric anisotropy, we consider the relationship is fairly straightforward since we can regard other properties (except for COF) as uninfluential factors, as answered in #1a and #1b.

4. Much of the discussion focuses on the detrended data. The manuscript mentions the method only briefly in the caption to Figure 3 and on Line 151. More description of the method, including

physical and statistical rationale for the choice and comparison with other methods, would provide more confidence in the value of the detrended data.

We will add following sentences at around Line 154:

*The detrended $\Delta\varepsilon$ represents the relative degree of c-axis clustering and the extent of deformation relative to the surrounding depth. Detrended values are more useful than original values to investigate the fluctuations of COF and to compare with other physicochemical properties.*

5. I suggest that a revised manuscript include more statistical exploration of the data comparisons stemming from Fig. 9. I noticed two locations with reported correlation coefficients (lines 355 and 390), which seem to be for timeseries pairs (e.g., delta-e and HCl). I feel that a more systematic, potentially multivariate approach would have more value. Part of this request is to add more reliability to the interpretations: at present, the mixed signals of whether dust or Cl or something else will affect delta-e (e.g., Type A and Type B relationships) does not provide a pathway to predict the effect.

The concentration of dust particles was measured by a laser particle counter. In this system, we can verify the increase of dust concentrations when the increasing amount is significantly large; however, it is difficult to detect the small variations of dust concentrations. Therefore, multivariate analysis using dust concentrations is difficult. Additionally, we could find correlations between dielectric anisotropy and concentrations of chloride ions and dust particles in only specific depths, so valuable results are not expected in a multivariate analysis through whole depth.

---

## Author Response (AR1)

Many thanks for big efforts by both of reviewers and editors for the manuscript tc-2021-336.

We have revised the manuscript according to the comments and suggestions from reviewers and editor. The grammar in changes and additions were checked by a native speaker of English.

The modifications (highlighted in yellow in the revised manuscript) are as follows:

**Added sentences, references, and figure**

[revised manuscript text omitted]

**Modification**

(Figure 3) We changed plot colors. (RC1)

(Conclusion) We simplified "Conclusion" section and used bullet. (RC1)

---

## Author Response (AR2)

Reply to the editor

We thank the editor for careful review of our manuscript and thoughtful comments to improve it. In the following, we describe our responses (in blue) point-by-point to each of editor's comment (in black).
In the marked-up manuscript, the modified (added) and removed words are highlighted in yellow and blue colors, respectively.

Line 16: change "… the present work investigated…" to "… the work presented here investigates…"
We modified.

Line 17: remove "depth within the 3035 m long"; remove "drilled at one of the dome summits"
We removed.

Line 19: it's not clear what the phrase "at which the ice cover has an age of approximately 300 kyrs BP." Perhaps you mean that the top 80% of the Dome Fuji ice core studied is ice from the last 300 kyr BP? I think you can remove that phrase and change the comma here to a period.
We removed.

Line 20: change "…moving in the depth direction" to "…with depth"
We modified.

Line 26: change "…together with..." to "…and…"
We modified.

Line 48: change "As an…" to "For…"
We modified.

Line 64: change "…and so…" to ", therefore"; change "efforts" to "effort"
We modified.

Lines 99-100: change to "At present, under the Holocene climate,…"
We modified.

Line 106: change to "The DF1 and DF2 boreholes are only 48 m apart."

We modified.

Lines 108-109: change to "from the depths of 100 to 2400 m."
The sentence was remodified by professional as follows:
"between the depths of 100 and 2400 m"

Line 109: change to "At each sampling depth,…"
We modified.

Line 160: change to "Error bars…"
We modified.

Line 164: change "(plot a)" to "(Fig. 4a)"; change "(plot b)" to "(Fig. 4b)"
We modified.

Line 165: add "Fig. 4c" to the parentheses here
We added.

Line 166: change "(plot d)" to "(Fig. 4d)"
We modified.

Line 171: change "Plot (a)" to "Figure 4a"; remove "is to"
We modified.

Lines 174-175: change "plot (a)" to "Fig. 4a"
We modified.

Line 175: double-check that "MIS7abc" is correct. Do you need commas between a, b, and c?
We don't need commas between abc. (checked by professionals)

Line 177: change "Plot (a)" to "Figure 4a"
We modified.

Line 181: change "plot (c)" to "Fig. 4c"
We modified.

Line 189: change to "Error bars…"
We modified.

Line 191: change "panel a" to "panel (a)"
We modified.

Line 208: change "… our present newly developed method." to "… our newly developed method that we present here."
We modified.

Line 211: change "panel (a)" to "Fig. 5a"
We modified.

Line 212: change "Panel (b)" to "Figure 5b"
We modified.

Line 286: add "transitions" after "interglacial to glacial"
We modified.

Line 291: change "discussions" to "discussion"
We modified.

Line 327: add "(Sect. 1)" after "Introduction"
We added.

Line 334: add "a" before "complex"
We added.

Line 337: remove "as to" after "(v)"; add "our" after "apply"
We modified.

Line 349: change "y-scale" to "y-axis"
We modified.

Line 370: capitalize the 't' so that it is defined as "Type A"
Line 373: capitalize the 't' so that it is defined as "Type B"

Line 377: capitalize the 't' in "Type"

Line 405: capitalize the 't' in "Type"

Line 409: capitalize the 't' in "Type"

Line 413: capitalize the 't' in "Type"

Line 417: capitalize the 't' in "Type"

Line 434: capitalize the 't' in "Type"

Line 451: capitalize the 't' in "Type"

Line 479: capitalize the 't' in "Type"

Line 495: capitalize the 't' in "Type" in both places here

We modified all.

Line 385: add "the" before "plots"

We added.

Line 390: change "so that" to "therefore"

We modified.

Line 419: add "is" before "slightly"

We added.

Line 420: change "reason for" to "cause of"

We modified.

Line 436: add "Type" before "B"

We added.

Line 444: change the period (.) to a colon (:) here; change "These are" to (i)

We modified.

Line 445: add "(ii)" before "the various…"

We modified.

Line 446: change to "In the first case, if the deformation…"

We modified.

Line 450: change to "In the second case, deformation…"

We modified.

Line 500: change "and so" to ", which is"
We modified.

Line 504: change to "near this bubble close-off horizon"
We modified.

Line 507: clarify where this phenomena was observed (in your samples for this study? Or in DF samples from Fujita et al?); change "and/or" to "as well as"
We added "Fujita et al., 2016" and modified.

Line 510: change to "metamorphism and deformation"
We modified.

Line 526: change to "the positive and negative…"
We modified.

Line 544: add "of the DF ice core" after "80%"; the phrase "which will be continuous to the deeper 20%" doesn't make sense. Consider removing that phrase.
We added and removed.

Line 546: change "contrasts" to "fluctuations"
We modified.

Line 547: change "stresses/strain" to "stress/strain"
We modified.

Line 549: change "contrasts" to "fluctuations"
We modified.

Line 552: change to "ice sheets"
We modified.

Line 570: change to "the presence of layers with heterogeneous thickness could…"
We modified.

Lines 571-574: it's not clear what you mean in these sentences. Please revise so that the reader can understand the point you are making here.

We modified the sentences as follows:

"*Finally, we suggest an important implication obtained from this study. Layered sequences of ice core signals resulting from Cl⁻ and dust particles are commonly obtained from inland ice core sites over the entire widths of the Antarctic ice sheet with minor local variations. Because both Cl⁻ and dust particles are primary factors controlling the development of COF (as demonstrated by this study), the COF layering profiles established by these factors should be similar in many ice core sites located in inland of the Antarctic ice sheet. Profiles of COF layering should be similar not only at ice core sites but also over much wider areas of the ice sheet having common sequences of both Cl⁻ and dust particles.*"

Line 574: add "a" after "discovered"
Line 575: add "the" before "EDC"; change to "The two sites are approximately 2,000 km apart in East Antarctica."
Line 576: change "for" to "of"
Line 577: change to "Since we can use radars…"; add "now" after "should"

The sentences including these points were remodified by professionals as follows:

"*Indeed, in Figure 8, the variations in $a_3^{(2)}$ eigenvalues over time in the DF2 ice core (this study) and the EDC ice core are seen to be similar even though these two sites are approximately 2000 km apart in East Antarctica. This is the first example of common features of COF variations within two very remote ice cores and is thus an important finding. Because radar data can be used to detect COF layering, it should be possible to compare deep COF layers across very wide areas of ice sheets, and such analyses should be performed at other locations.*"

Lines 580-581: change to "… we assessed the potential of using the dielectric anisotropy…"
We modified.

Line 584: add a comma here so that it is "Examining thick, 1-m long ice core specimens…"
We modified.

Line 589: change to "…values increased with increasing depth…"
The sentence was remodified by professionals as follows:
"*values was to increase with increasing depth*"

Line 590: add commas so that it is "The overall trend, in which the values increased, is…"

The sentence was remodified by professionals as follows:

*"This general pattern in which the values increase~"*

Lines 598-599: add commas so that it is "…Cl- ions, along with a negative correlation with the amount of dust particles in the ice core, were …"; consider changing "amount" to "concentration"

We added and modified.

Line 607: add comma after "sheets"

We added.

Lines 610-611: the phrase "…in layered manner were apparently present…" doesn't make sense. Please revise.

We removed "layered manner were apparently present"

Lines 612-615: this long sentence is also hard to understand; please revise. Remove both instances of "basically" in this sentence.

We modified this sentence as follows:

*"It should be emphasized that layered sequences of ice core signals related to Cl⁻ and dust particles were common at inland ice core sites within the wide Antarctic ice sheet. Because both Cl⁻ and dust particles are among the major factors determining the development of COF, profiles of COF layering established by these factors toward very deep depths should be similar within many ice core sites located in the inland regions of the Antarctic ice sheet and even in wider areas of the ice sheet having similar sequences of both Cl⁻ and dust particles."*

Line 617: add "the" after "within"

We added.

Line 618: add "the" before "future"

We added.

Line 619: change to "layers with heterogeneous thickness"

We modified.

Figure 5b: consider changing colors for the black markers (or to more visible markers), because it's very hard to tell the difference between the two black lines.

We modified to grey color

Figure 9 caption: include definitions for both the red and blue lines in the caption here.

We added definitions:

*Figure 9. Comparison of the S.D. values of $\Delta\varepsilon$ data determined at intervals of approximately 0.5 m along the core sample, using approximately 23 data points for each interval (blue line) and detrended $\Delta\varepsilon$ data defined as deviations from the third-order-fitting to the data in Fig. 4a (redlines) (smoothed over 10 m intervals). Note that the y-axis for the detrended $\Delta\varepsilon$ has been inverted.*

Figure 10 caption: define what the grey and tan shaded bars represent somewhere in the figure caption. Define what "AI", "AII", "B1", "A7", "B2", "AIII", and "B3" mean in the figure caption. Additionally, define what "AI", "AII", "B1", "A7", "B2", "AIII", and "B3" mean somewhere in the discussion section.

We added definitions of shaded bars and AI ~ B3 in the caption and body text (L371 and 375).

Figure 10 caption: "*Grey shading indicates correlations between the large depressions in the detrended $\Delta\varepsilon$ values and lower concentrations of soluble impurities (defined as Type A). These regions correspond to the termination-I/MIS1 (AI), termination-II/MIS5e (AII) and termination-III/MIS7e (AIII) transition periods and to MIS7abc (A7). Note that the labels indicate the termination or MIS stage in each case. Brown shading indicates correlations between the depressions in detrended $\Delta\varepsilon$ values and higher concentrations of dust (defined as Type B, numbered in order according increasing depth).*"

Non-public comments to the Author:

Many of the technical corrections listed above are due to grammar and sentence structure issues. The new portions of text you've added due to the reviewer comments have great grammar. Perhaps you could ask the same person to review the whole manuscript, as there might be more grammatical errors than the ones I found in the list above.

Thank you for careful review. The revised manuscript was checked by a native speaker.